# Interpretable Reinforcement Learning with Self-Abstraction and Refinement

## Abstract

We propose ReLIC, a reinforcement learning method with interactivity for composite tasks. Traditional RL methods lack interpretability, so it is difficult to integrate expert knowledge and refine the trained model. ReLIC is composed of a high-level logical model, low-level action policies, and a self-abstraction and refinement module. At its high level, it takes in predicates as its input so that we can design a synthesis algorithm to illustrate our high-level model's logical structure as an automaton, demonstrating our model's interpretability. At its low level, deep reinforcement learning is utilized for detailed action control to maintain high performance. Furthermore, based on the structured information provided by the automaton, ReLIC leverages GPT-4o to generate expert predicates and refine the automaton by injecting expert predicates and performing joint training, thereby enhancing RELIC's performance. ReLIC outperforms state-of-the-art baselines in several benchmarks with continuous state and action spaces. Additionally, ReLIC does not require humans to hard-code logical structures, so it can solve logically uncertain tasks.

## 1 Introduction

Although Reinforcement Learning (RL) has achieved tremendous success in a variety of control tasks, it still faces a significant challenge: lack of interpretability (Milani et al., 2022a). This deficiency manifests critically in two aspects: firstly, RL models often fail to provide transparent explanations for their decisions, posing significant risks in high-stakes domains such as autonomous driving (Song et al., 2022) and healthcare (Ahmad et al., 2018). Secondly, the absence of interpretability hinders direct interaction with models, making it difficult to integrate expert knowledge or intervene upon detecting biased or undesired behaviors (Rong et al., 2023).

To address the above challenges, researchers have proposed various interpretable RL methods. Early efforts emphasize *interpretability*, including decision tree (Bastani et al., 2018; Charbuty & Abdulazeez, 2021), programmatic policy (Trivedi et al., 2021), and Inductive Logic Programming (ILP)-based approaches Lavrac & Dzeroski (1994) such as NUDGE (Delfosse et al., 2023). However, these models prioritize transparency over interactivity and offer no interface for incorporating external knowledge. More recent work provides *interactivity* by integrating experts or Large Language Model (LLM), such as INTERPRETER (Kohler et al., 2024) and SCoBots (Delfosse et al., 2024). Nevertheless, existing interactive methods introduce expert or LLM knowledge after training, which means the final model is determined by human intuition, lacking dynamic integration of expert knowledge during the learning process.

In this paper, we propose **Reinforcement Learning with Interactivity for Composite tasks (ReLIC)** that integrates interpretability and lightweight, on-the-fly knowledge injection. ReLIC consists of modules on two levels: a lower-level module that executes concrete actions and an upper-level module that symbolically abstracts lower-level control logic. Specifically, the high-level logical model captures key runtime states through logical combinations of predicates. It has two notable advantages: unlike methods such as SCoBots (Delfosse et al., 2024), our logical model does not require pre-defined logical structures. Another feature is its ability to *incorporate additional expert knowledge* to guide training. To facilitate expert knowledge injection, we present a **self-abstraction and refinement** pipeline. In contrast to prior interactive methods (Kohler et al., 2024; Delfosse et al., 2024), our high-level logical model can condense a large predicate set into a compact set of salient

predicates. Next, our pipeline synthesizes an automaton from key predicates, clearly explaining the model's current behavior. Based on the synthesized automaton, an LLM then provides supplementary expert knowledge for the logic model, which realizes interactivity between the LLM and the logical model. As refinement alters the input predicates to the logical planner, we introduce **joint training** to co-optimize the logical planner and action policies, ensuring that injected knowledge seamlessly integrates into the policy execution.

Our contributions can be summarized as follows:

• **ReLIC Framework.** We propose a hierarchical RL framework that provides an interface to inject expert knowledge and supports joint training of the logical model and action policies.

• **Interpretability.** We introduce a self-abstraction technique that synthesizes an automaton from the logical model, providing a compact and transparent representation of learned behavior.

• **Interactivity.** We leverage an LLM to inject additional knowledge into the logical model based on the automaton, enabling dynamic updates in the learning process.

## 2 PRELIMINARIES

**Markov decision process (MDP):** MDP (Puterman, 1990) formalizes sequential decision-making under uncertainty as a tuple $(S, A, T, R, \gamma)$, where $S$ denotes state space, $A$ is the action space, $T$ is the transition function, $R$ is the reward function, and $\gamma$ is the discount factor. Given the current state $s_t$, the agent selects action $a_t$, transitions to $s_{t+1} \sim T(\cdot|s_t, a_t)$, and receives reward $r_{t+1} = R(s_t, a_t, s_{t+1})$. The objective is to find a policy $\pi$ that maximizes the expected cumulative reward over time. It is typically realized via the value function $V(s)$, which quantifies the expected return from state $s$ under policy $\pi$.

**First-order logic (FOL):** FOL (Barwise, 1977) is a formal language describing objects and their relations. It comprises constants, variables, predicates, and clauses. Constants denote specific objects in the environment, while variables represent unspecified ones. Predicates can be written as $P$, and an n-ary predicate is denoted as $P(x_1, x_2, ..., x_n)$, where $x$ represents constants or variables. Predicates capture properties or relations among objects, with truth values in true, false. A clause is a rule of the form $p_1 \leftarrow p_2, p_3, ..., p_n$, where $p_1$ is the head predicate and $p_2, p_3, ..., p_n$ are body predicates. Predicates grounded with constants are *extensional predicates* and serve as input predicates in our model; those defined via clauses are *intensional predicates* and correspond to target predicates.

**Differentiable Logic Machine (DLM):** DLM (Zimmer et al., 2021) is a trainable architecture for reasoning over predicates. It takes in a series of predicates as input and performs differentiable logic operations including *fuzzy and* $\wedge$, *fuzzy or* $\vee$, and *fuzzy not* $\neg$. DLM has a max depth of $D$. In each layer, it has $B$ computing units corresponding to predicates of different arities. The $b$-ary predicates in the $d$-th layer are denoted by $P_{d,b}(x_1, x_2, ...x_b)$, where $b \in \{1, 2, ..., B\}$. Each unit computes $b$-ary predicates and forwards them to the next layer. To enable computation across predicates of different arities, DLM employs *expansion* and *reduction* operations. Each unit applies binary operations over predicates from the previous layer with arities in $b-1, b, b+1$, using compositions such as: $P_{d,b} = P_{d-1,x} \wedge P_{d-1,y}$, $P_{d,b} = P_{d-1,x} \vee P_{d-1,y}$, $P_{d,b} = P_{d-1,x} \wedge \neg P_{d-1,y}$, $P_{d,b} = P_{d-1,x} \vee \neg P_{d-1,y}$, where $x, y \in \{b-1, b, b+1\}$.

## 3 METHOD

Our framework is shown in Figure 1. This framework assumes that an integrated task can be divided into many logically interdependent sub-tasks. Based on this assumption, the model comprises two levels: a *high-level logical model* for sub-task identification and planning, and a pool of *low-level action policies*, assumed to be pre-selected and pre-trained following previous work (Yang et al., 2020; He et al., 2022), responsible for executing individual sub-tasks. We first introduce each module of ReLIC in § 3.1. Next, we detail the joint training of logical planner and action policies in § 3.2. Finally, we present the self-abstraction and refinement process in § 3.3.

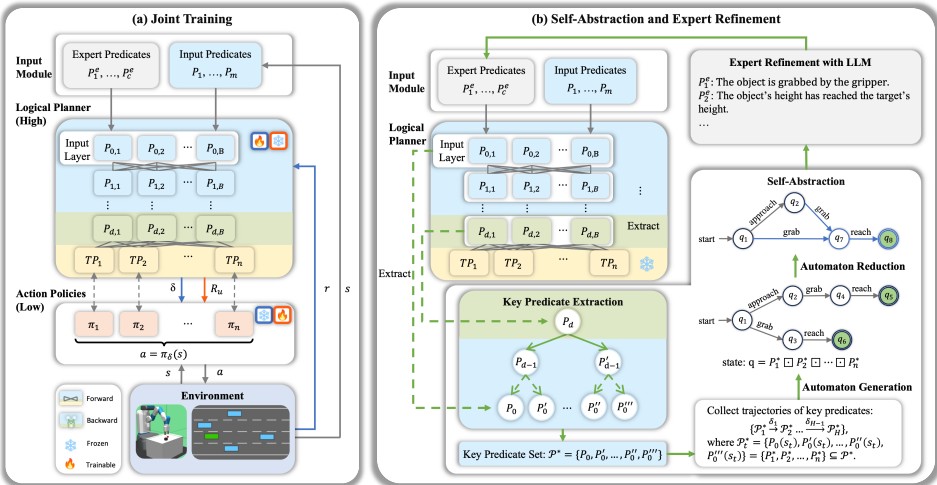

Figure 1: **The framework of ReLIC.** (a) **Joint Training (§ 3.2).** The *input module* first maps MDP states into input predicates, which, together with expert predicates (could be empty initially), form the *input* of the *logical planner*. The planner then produces target predicates (TP), each corresponding to an *action policy*, and samples an index $\delta$ (Eq. 2) to select the policy that acts in the environment. During logical model training (blue arrows and boxes), the planner samples $\delta$ and receives an environment reward $r$. Conversely, when training action policies (red elements), the planner remains fixed and provides a reward $R_u$ to guide policy updates. (b) **Self-Abstraction and Refinement (Green Arrows, § 3.3).** After recording the predicates from the planner's forward pass, *key predicate extraction* is performed via backpropagation to identify the *key predicate set* $\mathcal{P}^*$. Next, using trajectories of key predicates, an automaton is synthesized and simplified through *automaton generation* and *reduction*. This automaton is then *refined* by an LLM, producing expert predicates that update the input module. This process can be iterated to improve model performance.

## 3.1 ReLIC Framework

**Input module.** The input module transforms the MDP observations $s_t \in \mathbb{R}^K$ (at time $t$) into logical predicates that serve as the inputs of the high-level DLM. In general, a $b$-ary logical predicate $P$ is defined based on a $b$-ary real *transformation function* $f : \mathbb{R}^b \to \mathbb{R}$, a list of indices $i_1, i_2, \ldots, i_b \in [K]$, and an *activation interval* $(u, v)$. The predicate is then generated by $P(s_t) \leftarrow f((s_t)_{i_1}, (s_t)_{i_2}, \ldots, (s_t)_{i_b}) \in (u, v)$. The arity of a usual logical predicate used in our model is at most 3, which is enough for our experiments. We adopt the transformation functions in the simple forms of addition/subtraction so that they can be generally useful for most natural tasks. We list the adopted functions in the following table and note that functions such as $(x, y, z) \mapsto x + y - z$ can be substituted by $(x, y, z) \mapsto x - y + z$ via changing the order.

| Arity | Transformation functions |
|---|---|
| 1 | $x \mapsto x, x \mapsto |x|$ |
| 2 | $(x, y) \mapsto x + y, (x, y) \mapsto x - y$ |
| 3 | $(x, y, z) \mapsto x + y + z, (x, y, z) \mapsto x + y - z$ |

Going through the combinations of the transformation functions, the indices, and the activation intervals, the input module obtains a sequence of predicates and forms the set of *input predicates*:

$$\mathcal{P} = \{P_1, P_2, \ldots, P_m\}. \tag{1}$$

For instance, the predicate $P_1$ is generated by the transformation function $|x_{\text{obj}} - x_{\text{gripper}}| \in (0, 0.002)$, where $x_{\text{obj}}$ and $x_{\text{gripper}}$ are elements of the MDP observation $s$, and $(0, 0.002)$ denotes the activation interval. $P_1$ is true when the x-axis distance between the object and the gripper is less than 0.002, and false otherwise. Full lists of predicates appear in Appendix I.

Beyond basic conversion, the input module provides an interface to *inject expert knowledge*. Based on the understanding of a specific task, a human expert or LLM may introduce predicates with special transformation functions on particular indices of the MDP state vector, substantially improving

the planning performance. We define these predicates as *expert predicates* $P^e \in \mathcal{P}^e$, where $\mathcal{P}^e$ is the set of expert predicates. Please refer to § 4.3 for detailed demonstrations.

**High-level decision and the choice of an action policy.** Suppose there are $n$ low-level action policies. After the final layer (layer $d$) of the high-level DLM, we append a fully connected layer (layer $(d+1)$) so that there are $n$ special *target predicates*, $\mathrm{TP}_1, \mathrm{TP}_2, \ldots, \mathrm{TP}_n$, each of which corresponds to an action policy. An index $\delta_t$ is sampled via

$$\delta_t \sim \mathrm{softmax}\{\mathrm{TP}_1, \mathrm{TP}_2, \ldots, \mathrm{TP}_n\}. \tag{2}$$

**Deciding MDP actions.** Finally, we invoke the low-level action policy $\pi_\delta$ and take the MDP action:

$$a_t \leftarrow \pi_{\delta_t}(s_t). \tag{3}$$

**Key predicate extraction.** We recorded the predicates during the forward propagation of DLM, enabling the extraction of the key predicate. As outlined in § 2, DLM performs logical computations through the equation $P_d = (\sum w_{P_{d-1}} P_{d-1}) \boxdot (\sum w_{P'_{d-1}} P'_{d-1})$, where the operator $\boxdot \in \{\wedge, \vee, \wedge\neg, \vee\neg\}$ denotes logical conjunction or disjunction with optional negation. We extract the predicate $P_{d-1}$ and $P'_{d-1}$ with maximal weights $w_{P_{d-1}}$ and $w_{P'_{d-1}}$. We propagate this operation backward from the output layer to the input layer, decomposing the predicate in the $i^{th}$ layer with two predicates of the largest weight in the $(i-1)^{th}$ layer. Eventually, we define the extracted input-layer predicates as the *key predicate* $P^* \in \mathcal{P}^*$, where $\mathcal{P}^*$ is the set of key predicates.

**Automaton generation and refinement.** The extracted key predicates are used for the automaton synthesis algorithm in Appendix G. After getting an interpretable automaton, we refine it by injecting expert knowledge through special predicates provided by LLM, and integrate these predicates with input predicates and use them to train the high-level logical model. Details are in § 3.3.

## 3.2 JOINT TRAINING OF LOGICAL PLANNER AND ACTION POLICY

We propose a joint training framework (Figure 1(a)) that serves two key purposes. First, the high-level logical model and the pre-trained low-level policies require training. Second, after self-abstraction and refinement, the LLM injects additional expert knowledge, requiring further co-optimization of both modules to ensure that the injected knowledge is effectively propagated throughout the entire system. A highlight of our training algorithms is the surrogate rewards and training objectives, which crucially rely on our model structure and help to achieve superior performance.

---

**Algorithm 1** Joint Training Algorithm

1: **Input:** high-level DLM policy $\pi(\cdot|\theta_{\mathrm{DLM}})$ as described in Eq. (2), low-level action policies $\{\pi_i(\cdot|\theta_i)\}$, low-level critic $\{Q_i(\cdot|\theta_{Q_i})\}$, horizon $H$, volley size $\tau_{\mathrm{volley}}$, learning rate for actor $\alpha$, learning rate for critic $\beta$
2: $t \leftarrow 0$, observe the environment state $s_0$
3: **while** task not completed and $t < H$ **do**
4:     calculate the input predicates $\mathcal{P}$ based on $s_t$
5:     sample an index $\delta_t \sim \pi(P(s_t)|\theta_{\mathrm{DLM}})$
6:     **for** $j \leftarrow 1$ to $\tau_{\mathrm{volley}}$ **do**
7:         obtain action $a_{t+j}$ from $\pi_{\delta_t}(s_{t+j}|\theta_{\delta_t})$
8:         receive the reward $r_{t+j}$ from environment, observe the new environment state $s_{t+j+1}$
9:         get the estimated value $\omega_{t+j}$ from the critic network of DLM
10:        compute $R_u(r_{t+j}, \delta_t, \delta_{t+j}, \omega_{t+j-1}, \omega_{t+j})$
11:        $\theta_{\delta_t} \leftarrow \nabla(\log \pi_{\theta_{\delta_t}}(a_{t+j}|s_{t+j-1}) Q_{\delta_t}(s_{t+j-1}, a_{t+j-1})) + \alpha \cdot \theta_{\delta_t}$
12:        $\theta_{Q_{\delta_t}} \leftarrow \nabla(R_u + \gamma Q_{\delta_t}(s_{t+j}, a_{t+j}) - Q_{\delta_t}(s_{t+j-1}, a_{t+j-1})) + \beta \cdot \theta_{Q_{\delta_t}}$
13:     **end for**
14:     $\theta_{\mathrm{DLM}} \leftarrow \nabla(\log \pi_{\theta_{\mathrm{DLM}}}(\delta'_t|P(s_v)))(\sum_{v=0}^{T/\tau_{\mathrm{volley}}} \gamma^v r'_v) + \alpha \cdot \theta_{\mathrm{DLM}}$
15:     $t \leftarrow t + \tau_{\mathrm{volley}}$
16: **end while**

---

**Training the high-level logical model.** Here we illustrate how to train the high-level logical model while fixing the low-level policies. Inspired by Bacon et al. (2017b), we adopt a *volley*-based approach to address the common challenge of sparse environment rewards in RL algorithms

(Mnih et al., 2013). Specifically, when the high-level model selects a low-level action policy $\pi_\delta$, the chosen policy is executed for multiple consecutive steps (a *volley*), rather than a single step. The environment rewards collected during this volley are aggregated into *volley rewards*, which are used to train the high-level model. In Algorithm 1 line 3 to line 15, we roll-out the trajectory based on volleys: $\{(s'_v, \delta'_v, r'_v)\}_{v \in \{0,1,2,...\}}$. *These shorter volley-based trajectories feature denser reward signals*, improving sample efficiency. We apply the standard PPO algorithm (Schulman et al., 2017) to the volley-based trajectory to optimize the high-level policy $\pi(\cdot|\theta_{\mathrm{DLM}})$. In PPO, we also train a neural network fed by the input predicates as the critic to approximate the value function.

**Training low-level action policies.** With the high-level logical model fixed, low-level action policies are trained via policy gradient methods (e.g., DDPG (Lillicrap et al., 2015)). During training, complete task trajectories are rolled out and segmented according to the selected low-level policy. Each policy is then independently updated via gradient descent on its respective trajectory segments. The (DDPG-based) roll-out algorithm is described in Algorithm 1 line 6 to line 13. A critical component is the surrogate reward, defined as:

$$R_u(r, \delta, \delta', \omega, \omega') = r + \alpha \cdot 1[\delta \neq \delta' \wedge \omega > \omega'], \tag{4}$$

where $\alpha > 0$ is a hyperparameter. The idea of the surrogate reward is to integrate the environmental reward $r$, the instruction $\delta$ from the high-level model, and the estimated value $\omega$ from the DLM critic. When the current sub-task is completed, the high-level model switches to a new action policy ($\delta \neq \delta'$) and the expected value increases ($\omega > \omega'$). Therefore, adding the term $\alpha \cdot 1[\delta \neq \delta' \wedge \omega > \omega']$ incentivizes the current low-level action policy to learn to complete the sub-task requested by the high-level model. We provide the proof for this reward function in Appendix B.

**Joint training.** During the actual training process, we alternate between optimizing the high-level logical model and the low-level action policies, keeping one component's parameters fixed while updating the other. This alternating approach allows the two modules to progressively refine each other, enhancing coordination and guiding the system toward a jointly optimal solution.

## 3.3 SELF-ABSTRACTION AND REFINEMENT

Figure 1(b) illustrates our self-abstraction and refinement pipeline. Compared to prior work (Kohler et al., 2024; Delfosse et al., 2024), our pipeline offers two principal advantages. First, we employ the automaton that condenses trajectories of key predicates into compact states, without relying on predefined logical structures. Second, LLM can refine the automaton by injecting additional expert knowledge. These properties make our pipeline more interpretable and more interactive than prior methods. In the rest of this section, we describe how to perform self-abstraction through automaton synthesis in § 3.3.1, followed by expert refinement via LLM in § 3.3.2.

### 3.3.1 AUTOMATON GENERATION AND REDUCTION

We perform self-abstraction by synthesizing a Deterministic Finite Automaton (DFA) that abstracts the complex logical structures learned by the high-level model. We further evaluate the correctness of the synthesized DFA. The detailed algorithm for automaton synthesis is depicted in Appendix G.

First of all, we extract the *key predicate* after joint training. This approach allows us to focus on the input predicates that truly impact the decision-making process and groups the observations from the environment into a limited number of automaton states. Secondly, we utilize the ReLIC to track changes in the high-level model's decision. We run the ReLIC and record the bool value of the key predicate $P^*$ and the new decision $\delta$ every time the high-level model's output decision changes. The $q$ and $\delta$ are defined as the state and transition edge of the DFA:

$$q = P_1^* \boxdot P_2^* \boxdot ... \boxdot P_n^*, \tag{5}$$

where $P_i^* \in \mathcal{P}^*$ and $\boxdot \in \{\wedge, \vee, \wedge\neg, \vee\neg\}$. With this approach, each run yields a path in the automaton. We merge the nodes with the same $P^*$ and the edges with the same $\delta$ and prior node to get a complicated automaton. Finally, we apply the Hopcroft Algorithm (Gries, 1973) to reduce the automaton.

### 3.3.2 EXPERT REFINEMENT VIA LLM

The input for the high-level model is composed of all the predicates in $\mathcal{P}$ (Eq. 1), while the synthesized automaton in §3.3.1 keeps only key predicates $P^*$ for representing abstract states. Thus, it

might miss some critical predicates. We aim to make the abstraction more fine-grained by *integrating expert knowledge* through LLM-based refinement. We employ OpenAI's GPT-4o to perform expert refinement on the automaton because of its strong ability to analyze structured information and follow instructions.[1] We *input* the synthesized automaton alongside two randomly sampled failure trajectories from the environment, instructing the LLM to identify if any key predicates are missing in the automaton's state and then add missing predicates through prompting. Our prompt enables the LLM to analyze the problem in a chain-of-thought manner (Wei et al., 2022). The analysis proceeds in the following steps: (a) leverage expert knowledge while learning the analytical logic based on the input automaton, (b) analyze the logical relations between key predicates of states, (c) diagnose failure reasons for failed trajectories, (d) refine the automaton by proposing new expert predicates $P^e \in \mathcal{P}^e$, and (e) examine the explanation and refined automaton to form a conclusion. These steps allow the LLM to iteratively inspect its reasoning process and resulting outcomes. Then, we expand the set of key predicates $\mathcal{P}^*$ by $\mathcal{P}^* \leftarrow \mathcal{P}^* \cup \mathcal{P}^e$. We include the newly added predicates $\mathcal{P}^e$ to the input of the high-level logic model and perform the joint training algorithm to further fine-tune both the logical model and action policies. We repeat the process of joint training and adding new expert predicates several times until the training process finishes. The complete prompt template is provided in Appendix J.

## 4 EXPERIMENT

Our experiments aim to: (1) evaluate our method in comparison to baselines in challenging autonomous control environments (§ 4.2), (2) showcase the **interpretability** and **interactivity** gained through self-abstraction and refinement (§ 4.3), and (3) showcase the effectiveness of the automaton representation and the self-abstraction and refinement module (§ 4.4).

### 4.1 EXPERIMENTAL SETUP

**Highway environment.** *Highway* is an autonomous driving simulator based on the OpenAI Gym (Leurent, 2018). The objective is to control an ego vehicle to maintain high speed while avoiding collisions. The state space is $\mathbb{R}^{25}$, representing the ego vehicle and its four nearest surrounding vehicles. Each vehicle is characterized by five features: a binary existence flag, x and y positions, and x and y velocities. The action space is $\mathbb{R}^2$, encoding the ego vehicle's horizontal and angular accelerations.

**Fetch environment.** *Fetch-Pick-And-Place* in OpenAI Gym consists of a robotic arm and an object (Plappert et al., 2018). The task requires the robot to pick an object and place it at a specified position. The state space is $\mathbb{R}^{25}$, comprising the information of gripper, object, and target. The action space is $\mathbb{R}^4$: the first three dimensions encode the target gripper position and the fourth controls the gripper width. The object's initial position on the table is randomly generalized. We design four tasks to evaluate our model. *Pick&Place*: Pick the object and move it to a target position. *Pick&PlaceCorner*: Pick and lift the object, then move it to the top-right corner. *PickLiftPlace*: Pick and lift the object, then move it to a designated target position. *PickHighPlace*: Pick and lift the object to a high position, then place it at a target position, which may vary in height.

**Implementation details.** We train our model with 500 epochs and perform the expert refinement every 50 epochs. In each epoch, we have 8 episodes with horizon $H = 100$. In joint training, we set the volley length $\tau_{volley}$ as 10. For testing, we conduct our experiments using 10 random seeds, with each seed evaluated over 100 runs. Please check Appendix H for more implementation details.

**Baselines.** We compare ReLIC against representative baselines spanning four categories: **standard RLs**, **interpretable RLs**, **hierarchical RLs**, and **interactive RLs**. TD3-HER (Balasubramanian, 2023) represents standard RL. DiRL (Jothimurugan et al., 2021b) is a hierarchical RL method. Interpretable RL methods include NLM (Dong et al., 2019), DLM (Zimmer et al., 2021), NUDGE (Delfosse et al., 2023), and INSIGHT (Luo et al., 2024), where INSIGHT is an end-to-end neural-symbolic model, and NUDGE employs neural-guided symbolic abstraction. For interactive RL, we evaluate against SCoBots(Delfosse et al., 2024) and INTERPRETER(Kohler et al., 2024). For all discrete models (NLM, DLM, NUDGE, SCoBots, INTERPRETER), we provide the same low-level policies as ReLIC to ensure a fair comparison.

---

[1]https://platform.openai.com/docs/models/gpt-4o

Table 1: **Performance comparison on *Highway*.** ReLIC surpasses all baselines across standard RL (TD3-HER), hierarchical RL (DiRL), interpretable RL (NLM, DLM, NUDGE, and INSIGHT), and interactive RL (SCoBots and INTERPRETER). *Crash rate* denotes the percentage of episodes ending in failure ($\downarrow$ is better); *velocity* is the agent's average speed; and *length* measures how long the agent remains active ($\uparrow$ is better).

| | TD3-HER | DiRL | NLM | DLM | NUDGE | INSIGHT | SCoBots | INTERP. | ReLIC |
|---|---|---|---|---|---|---|---|---|---|
| length $\uparrow$ | $28.5_{\pm 0.1}$ | $45.6_{\pm 0.2}$ | $55.8_{\pm 0.3}$ | $56.8_{\pm 0.2}$ | $93.3_{\pm 0.1}$ | $34.6_{\pm 0.3}$ | $93.4_{\pm 0.3}$ | $95.5_{\pm 0.2}$ | $\mathbf{97.1_{\pm 0.1}}$ |
| velocity $\uparrow$ | $15.3_{\pm 0.2}$ | $26.5_{\pm 0.2}$ | $25.2_{\pm 0.2}$ | $26.4_{\pm 0.3}$ | $24.9_{\pm 0.2}$ | $20.1_{\pm 0.3}$ | $22.2_{\pm 0.3}$ | $25.7_{\pm 0.1}$ | $\mathbf{27.9_{\pm 0.1}}$ |
| crash rate (%) $\downarrow$ | $73.3_{\pm 2.0}$ | $76.1_{\pm 1.0}$ | $56.4_{\pm 0.8}$ | $54.3_{\pm 1.0}$ | $4.7_{\pm 1.0}$ | $70.7_{\pm 2.9}$ | $7.2_{\pm 0.9}$ | $5.4_{\pm 1.4}$ | $\mathbf{4.0_{\pm 0.8}}$ |

Table 2: **Performance comparison on *Fetch*.** ReLIC outperforms standard RL (TD3-HER), hierarchical RL (DiRL), interpretable RL (NLM, DLM, NUDGE, and INSIGHT), and interactive RL (SCoBots and INTERPRETER) baselines.

| Success rate (%) | TD3-HER | DiRL | NLM | DLM | NUDGE | INSIGHT | SCoBots | INTERP. | ReLIC |
|---|---|---|---|---|---|---|---|---|---|
| Pick&Place | $51.3_{\pm 1.7}$ | $93.7_{\pm 1.2}$ | $63.5_{\pm 2.3}$ | $73.2_{\pm 0.7}$ | $75.5_{\pm 2.0}$ | $51.6_{\pm 1.5}$ | $85.6_{\pm 1.4}$ | $90.0_{\pm 0.9}$ | $\mathbf{97.3_{\pm 0.9}}$ |
| Pick&PlaceCorner | $52.9_{\pm 2.1}$ | $93.1_{\pm 0.8}$ | $62.1_{\pm 1.5}$ | $68.5_{\pm 1.3}$ | $74.8_{\pm 1.2}$ | $54.4_{\pm 0.8}$ | $84.5_{\pm 1.0}$ | $88.6_{\pm 1.1}$ | $\mathbf{99.3_{\pm 0.8}}$ |
| PickLiftPlace | $50.0_{\pm 1.2}$ | $91.8_{\pm 1.2}$ | $59.7_{\pm 0.9}$ | $67.9_{\pm 0.8}$ | $73.2_{\pm 1.2}$ | $49.1_{\pm 1.9}$ | $84.7_{\pm 1.3}$ | $87.1_{\pm 1.1}$ | $\mathbf{99.0_{\pm 0.9}}$ |
| PickHighPlace | $21.5_{\pm 0.9}$ | $42.5_{\pm 0.7}$ | $30.3_{\pm 1.5}$ | $29.1_{\pm 0.5}$ | $40.9_{\pm 0.8}$ | $32.1_{\pm 0.8}$ | $76.8_{\pm 1.1}$ | $75.3_{\pm 1.5}$ | $\mathbf{90.5_{\pm 1.1}}$ |

## 4.2 MAIN RESULTS

Tables 1 and 2 present the performance of various methods on the *Highway* and *Fetch* environments, respectively. *Overall, ReLIC surpasses all baselines in all tasks.* TD3-HER, as a purely neural baseline, yields the weakest performance. Among interpretable RL methods, INSIGHT underperforms due to its neural policy and the lack of training feedback from its explanations. NLM, DLM, and NUDGE perform moderately well on *Fetch*, where the action space is small and task sub-structure is clear, but struggle with execution precision–for instance, NUDGE learns the correct high-level sequence in *PickLiftPlace* (approach then grab), but an incomplete approach may lead to grabbing failures. On the more dynamic *Highway* task, these methods perform poorly, even with provided neural low-level policies, as they lack *joint training* to coordinate high-level planning with low-level execution. DiRL, as a hierarchical RL method, performs well on *Fetch*, but fails on *Highway*, where the unpredictable behavior of other vehicles invalidates its fixed, predefined task hierarchy. Interactive baselines such as SCoBots and INTERPRETER benefit from expert or LLM guidance but lack mechanisms for iterative refinement and joint training, limiting further improvement. In contrast, ReLIC achieves superior performance by dynamically integrating LLM-provided predicates through *expert refinement* and *joint training*, effectively combining expert knowledge with model learning.

## 4.3 CASE STUDY: SELF-ABSTRACTION AND LLM REFINEMENT ON *PickHighPlace*

*PickHighPlace* is a challenging task because lifting the object to a high position increases the gripper's travel distance and makes the coherent action between the reach and lift sub-tasks difficult to learn. In this section, we select *PickHighPlace* as a case study to illustrate our self-abstraction and refinement. Another case study on *Highway* is in Appendix E.

**Automaton-based abstraction makes ReLIC interpretable.** Figure 2(a) demonstrates sampled trajectories of key predicates, where each environment state is mapped to a set of key predicates. These trajectories are condensed into an automaton, with transitions aligned to the low-level action policies `approach`, `grab`, and `reach`. The resulting automaton is depicted in Figure 2(b). To simplify it, we apply Hopcroft's algorithm to merge redundant or equivalent states; the reduced automaton appears in Figure 2(c). In this reduced version, state $q_7$ merges $q_3$ and $q_4$, representing the gripper grabs the object directly or after approaching, while state $q_8$ merges $q_5$ and $q_6$ to represent the final state. The resulting automaton's concise states and transitions offer an immediate interpretation of the agent's policy. For instance, state $q_1$ indicates that the gripper is near or far from the object, so only `approach` and `grab` are available. Choosing `approach` deterministically moves the agent to $q_2$, where the gripper is near the object and its opening exceeds the object's width.

**ReLIC agents are interactive via LLM refinement.** The imperfect learning status of the high-level logical model results in a decreased success rate when the lifting threshold is high. To ad-

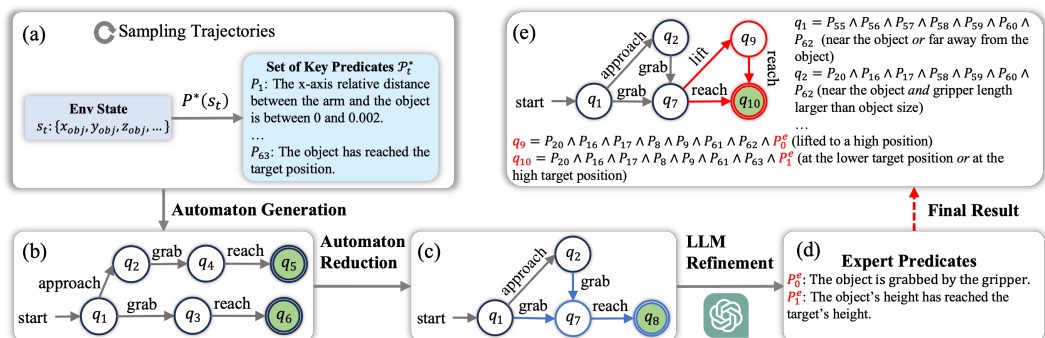

Figure 2: **Examples for self-abstraction and refinement for *PickHighPlace*. (a) Sampled trajectories of key predicates.** From environment state $s_t$, the key predicate set $\mathcal{P}_t^*$ is extracted, capturing critical symbolic conditions. **(b) Automaton generated from trajectories of key predicates.** Transitions between states are derived based on trajectories of key predicates. **(c) Automaton reduction.** Redundant or equivalent states are merged; blue edges and nodes represent updated transitions after reduction. **(d) LLM refinement.** The language model proposes additional predicates that encode expert knowledge. **(e) Final refined automaton.** Red edges and states are synthesized after another round of the whole process. More details are provided in Appendix F.

dress this, the LLM analyzes the reduced automaton and the sampled trajectories to identify missing environmental cues in the existing set of key predicates. It then proposes additional expert predicates to enrich the input. As shown in Figure 2(d), the LLM combines five original input predicates $P_1$: $|x_{\text{obj}} - x_{\text{gripper}}| \in (0, 0.002)$, $P_2$: $|y_{\text{obj}} - y_{\text{gripper}}| \in (0, 0.002)$, $P_3$: $|z_{\text{obj}} - z_{\text{gripper}}| \in (0, 0.002)$, $P_{19}$: $|d_0| \in (0.006, 0.008)$, $P_{20}$: $|d_1| \in (0.006, 0.008)$ to generate two expert predicates:

$$P_0^e: P_1 + P_2 + P_3 + |P_{19} + P_{20} - t| \in (0, err) \text{ and } P_1^e: |z_{\text{obj}} - 0.45| \in (0, err),$$

where $t$ is a gripper-gap threshold, $d_0$ and $d_1$ are the respective displacements of the left and right grippers, and $err$ is a tolerance parameter. $P_0^e$ indicates whether the gripper can successfully grasp the object, while $P_1^e$ checks whether the object has been lifted to the target height (0.45).

**Injected knowledge yields a better final automaton.** Figure 2(e) shows the automaton after injecting expert knowledge into the logical model and re-running the full pipeline. Two new states, $q_9$ and $q_{10}$, derived from state $q_8$ of the reduced automaton by incorporating LLM-generated expert predicates $P_0^e$ and $P_1^e$, respectively. These expert predicates make the automaton states more fine-grained, enabling the model to identify if the `lift` action is finished and execute `lift` and `reach` actions in proper sequence. As a result, ReLIC surpasses all baselines on *PickHighPlace*, as reported in Table 2.

## 4.4 ABLATION STUDY

We conduct three ablations to highlight the benefits of our automaton-structured input and LLM refinement. First, we compare ReLIC to a variant that removes the automaton and feeds trajectories of key predicates directly to the LLM (**w/o SA**). Second, we retain the automaton but replace LLM-generated expert predicates with manually crafted ones of the same logical form (**w/ HR**). Third, we disable the self-abstraction and refinement pipeline entirely, and keep joint training with the original predicate set (**w/o SAR**). Results for all variants are summarized in Table 3.

On the *PickHighPlace* task, ReLIC improves success rate by roughly 63% over the **w/o SA** variant. This gain stems from the self-abstraction stage: raw trajectories of key predicates are condensed into an automaton and further simplified, yielding a compact set of structured information. The resulting representation makes it easier for the LLM to spot missing information and generate precise expert predicates, thereby boosting performance. **ReLIC is competitive with a human expert** (**w/ HR**). The input space is large–64 predicates in *Fetch* and 88 in *Highway*–so even a human expert finds it difficult to identify the most relevant input predicates for constructing expert rules. When the self-abstraction and refinement pipeline is removed (**w/o SAR**), performance drops sharply–e.g., from 90% to 40% on *PickHighPlace*. This verifies that iterative self-abstraction and refinement are critical for uncovering missing state cues and guiding joint training toward task success.

Table 3: **Self-abstraction and refinement significantly enhance performance.** *w/o SA*: Without self-abstraction; *w/o SAR*: Without self-abstraction and refinement; *w/ HR*: With human refinement.

| Task | Metric | ReLIC | w/o SA | w/o SAR | w/ HR |
|---|---|---|---|---|---|
| Pick&Place | | $\mathbf{97.3}_{\pm\mathbf{0.9}}$ | $93.6_{\pm1.3}$ | $86.2_{\pm1.5}$ | $97.0_{\pm1.3}$ |
| Pick&PlaceCorner | Success rate (%) | $\mathbf{99.3}_{\pm\mathbf{0.8}}$ | $94.0_{\pm0.8}$ | $85.3_{\pm2.3}$ | $97.1_{\pm0.8}$ |
| PickLiftPlace | | $\mathbf{99.0}_{\pm\mathbf{0.9}}$ | $93.1_{\pm1.4}$ | $83.5_{\pm1.3}$ | $96.2_{\pm1.2}$ |
| PickHighPlace | | $\mathbf{90.5}_{\pm\mathbf{1.1}}$ | $55.2_{\pm1.7}$ | $40.0_{\pm1.8}$ | $88.6_{\pm0.8}$ |
| Highway | Length ($\uparrow$) | $\mathbf{97.1}_{\pm\mathbf{0.1}}$ | $94.8_{\pm0.1}$ | $91.4_{\pm0.2}$ | $96.6_{\pm0.1}$ |
| | Velocity ($\uparrow$) | $\mathbf{27.9}_{\pm\mathbf{0.1}}$ | $26.7_{\pm0.2}$ | $27.0_{\pm0.1}$ | $26.3_{\pm0.1}$ |
| | Crash rate (%) ($\downarrow$) | $\mathbf{4.0}_{\pm\mathbf{0.8}}$ | $4.1_{\pm1.0}$ | $5.3_{\pm0.6}$ | $4.0_{\pm1.1}$ |

## 5 RELATED WORK

**Hierarchical RL.** Early frameworks like Hierarchical Abstract Machines (HAMs) (Parr & Russell, 1997) provided designer-specified state-machine subroutines, while the option-critic architecture (Bacon et al., 2017a) learned temporally-extended actions end-to-end. Feudal Networks (Vezhnevets et al., 2017) introduced an explicit manager-worker hierarchy, with a high-level module setting latent subgoals for a low-level controller. More recently, DiRL (Jothimurugan et al., 2021a) used logical task specifications to automatically construct task graphs and learn a policy for each edge (subtask) with integrated high-level planning. ReLIC also leverages hierarchy, derives an automaton-based task structure through self-abstraction and refines it during learning, rather than relying on fixed or manually defined logical schemas.

**Interpretable RL.** Early work achieved interpretability by constraining the policy class to transparent structures such as decision trees (Bastani et al., 2018; Topin et al., 2021; Charbuty & Abdulazeez, 2021; Milani et al., 2022b; Kohler et al., 2024), graphs (Topin & Veloso, 2019), logical programs (Verma et al., 2018; 2019; Silver et al., 2020; Inala et al., 2020b), or state machines (Inala et al., 2020a). Post-processing interpretable models, such as Local Interpretable Model-agnostic Explanations (LIME) (Ribeiro et al., 2016; Zhao et al., 2021), Shapley-based methods (Kumar et al., 2020), and LLM-based methods (Luo et al., 2024), explain opaque agents after policy learning. Other works focus on learning interpretable logic policies via Inductive Logic Programming (ILP) (Lavrac & Dzeroski, 1994), which extracts rules from predefined templates. ILP struggles to scale up to complex scenarios because its rule space grows exponentially with task complexity (Cropper et al., 2022). Neural Logic Machines (NLM) (Dong et al., 2019) address this by introducing MLPs to improve expressiveness at the cost of transparency; Differentiable Logic Machines (DLM) (Zimmer et al., 2021) replace MLPs with fuzzy logic to restore readability. NUDGE (Delfosse et al., 2023) leverages trained neural agents to guide the search for candidate-weighted logic rules, thus providing interpretable policies. ReLIC differs from these methods by utilizing the logic machine as a high-level policy to facilitate expert knowledge injection.

**Interactive RL.** There is growing interest in interactive agents that can incorporate external knowledge. SCoBots (Delfosse et al., 2024) employ a priori concept bottlenecks to train agents that allow user inspection and intervention. INTERPRETER (Kohler et al., 2024) distills black-box policies into an editable program, allowing post hoc expert modification. BlendRL (Shindo et al., 2025) blends logic and neural policies using LLM to enhance performance. ReLIC distinguishes itself by joint training with external knowledge, integrating LLMs' feedback into the learning process to dynamically align the agent's behavior with expert intentions.

## 6 CONCLUSION

We introduce **ReLIC**, the Reinforcement Learning with Interactivity for Composite tasks. It integrates a logical model for high-level decision-making and symbolic abstraction, along with low-level action policies designed for the precise execution of sub-tasks. Notably, ReLIC excels in control tasks due to its **interpretability** and **interactivity**. Based on the structured information provided by the automaton, ReLIC utilizes LLM to provide interpretation, generate expert predicates, and perform self-refinement by injecting expert predicates and joint training. Despite these strengths, ReLIC still has the limitation of requiring pretrained low-level policies–a form of built-in expert knowledge. For future work, we aim to construct an RL system with minimal expert knowledge, which offers a promising avenue toward establishing a lifelong learning system.

ETHICS STATEMENT

This work adheres to the ICLR Code of Ethics. In this study, no human subjects or animal experimentation were involved. All environments used, including the Fetch and Highway, were sourced in compliance with relevant usage guidelines, ensuring no violation of privacy. We have taken care to avoid any biases or discriminatory outcomes in our research process. No personally identifiable information was used, and no experiments were conducted that could raise privacy or security concerns. We are committed to maintaining transparency and integrity throughout the research process.

REPRODUCIBILITY STATEMENT

We have made every effort to ensure that the results presented in this paper are reproducible. All code and datasets have been made publicly available in an anonymous repository to facilitate replication and verification. The experimental setup, including training steps, model configurations, and hardware details, is described in detail in the paper. We have also provided a full description of our ReLIC framework to assist others in reproducing our experiments. We believe these measures will enable other researchers to reproduce our work and further advance the field.

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

# A   APPENDIX

# B   PROOF FOR THE REWARD FUNCTION $R_u$

Here we show that the learned policy $\pi'(\cdot|\theta)$ under our transform of the reward function is included in the original optimal policy. Our proof follows the idea of the work (Ng et al., 1999). Since we use a method based on Q-learning for policy optimization, we have:

$$Q(s,a) = \mathbb{E}_s(R(s,a,s') + \gamma max_{a' \in A}Q(s',a')). \tag{6}$$

If we add a potential function $\Phi(s)$, which is only related to the states, to both sides of the equation, then:

$$Q(s,a) - \Phi(s) = \mathbb{E}_s(R(s,a,s') + \gamma\Phi(s') - \Phi(s) + \gamma(max_{a' \in A}Q(s',a') - \Phi(s'))). \tag{7}$$

Here we do a transformation to both the $Q$ function and the $R$,

$$Q'(s,a) = Q(s,a) - \Phi(s), \tag{8}$$
$$R_u(s,a,s') = R(s,a,s') + \gamma\Phi(s') - \Phi(s). \tag{9}$$

by substituting Eq. 7 with $Q'$ and $R_u$, we get the new formula

$$Q'(s,a) = \mathbb{E}_s(R_u(s,a,s') + \gamma max_{a' \in A}Q'(s',a')), \tag{10}$$

which keeps the form of Q-learning.

For our specific case, we let

$$\Phi(s) = \begin{cases} 1/\gamma & \text{if } \delta(s) \neq \delta(s') \wedge \omega(s) < \omega(s'), \\ 0 & otherwise. \end{cases} \tag{11}$$

Notice that $\delta(s)$ is the decision made by the logical model, so it is only related to state $s$. $\omega$ is the output of the high-level critic, though it is related to both the state $s$ and action $\delta$, in every volley, the decision of the high-level logical model remains the same, which means $\delta$ is always the same. So the $\omega$ is also just related to state $s$. Then the final form of our reward function is

$$R_u(r,\delta,\delta',\omega,\omega') = r + \alpha \times 1[\delta \neq \delta' \wedge \omega < \omega']. \tag{12}$$

# C   ADDITIONAL EXPERIMENT RESULTS

## C.1   ADAPTIVENESS

To show the adaptiveness of our ReLIC in tasks that are conceptually analogous yet distinct in their details, we fine-tune the model on the modified environment, whose side length of the cubic object is reduced from 0.25cm to 0.15cm. **It is worth noting that we only fine-tune a certain set of lower-level action policies** using Algorithm 3 while keeping the high-level logical model unchanged. The main experiment results are in § 4.

Table 4: **Performance comparison before and after fine-tuning after changing the object size.**

| Succ rate(%) | Pre-Finetune | Post-Finetune |
| --- | --- | --- |
| Pick&Place | $84.5_{\pm 1.6}$ | $\mathbf{94.3_{\pm 1.1}}$ |
| Pick&PlaceCorner | $85.8_{\pm 2.0}$ | $\mathbf{95.7_{\pm 1.2}}$ |
| PickLiftPlace | $84.0_{\pm 1.5}$ | $\mathbf{93.1_{\pm 0.7}}$ |

**Result.**   The experiment results are depicted in Table 4. After adjusting the size of the object, the success rate of the model decreases by around 10%. However, by solely fine-tuning the lower-level action policies, we can effectively recover the success rate lost. Additionally, since the lower-level action policy is relatively simple and has a low training cost, it signifies that we can quickly fine-tune our model to adapt to changes in the environment and task requirements.

# D    DETAILS OF JOINT TRAINING ALGORITHM

We provide the detailed algorithm for training the high-level logical model in Algorithm 2. Besides, we provide the detailed algorithm for training low-level action policies in Algorithm 3.

---

**Algorithm 2** Volley-based Roll-out for High-level Logical Model Training

---

1: **Input:** high-level DLM policy $\pi(\cdot|\theta_{\mathrm{DLM}})$ as described in Eq. (2), low-level action policies $\{\pi_i(\cdot|\theta_i)\}$, horizon $H$, volley size $\tau_{\mathrm{volley}}$
2: Volley count $v \leftarrow 0$
3: **while** $v < H/\tau_{\mathrm{volley}}$ **do**
4:     observe the environment state $s_{v \cdot \tau_{\mathrm{volley}}}$, let $s'_v \leftarrow s_{v \cdot \tau_{\mathrm{volley}}}$
5:     calculate the input predicates $\mathcal{P}$ based on $s'_v$, sample an index $\delta'_v \sim \pi(s'_v|\theta_{\mathrm{DLM}})$
6:     Volley reward $r'_v \leftarrow 0$
7:     **for** $j \leftarrow 0$ **to** $\tau_{\mathrm{volley}}$ **do**
8:         **if** $j \neq 0$ then observe the environment state $s_{v \cdot \tau_{\mathrm{volley}}+j}$ **then**
9:             execute the environment action $a_{v \cdot \tau_{\mathrm{volley}}+j} \leftarrow \pi_{\delta'_v}(s_{v \cdot \tau_{\mathrm{volley}}+j}|\theta_{\delta'_v})$, receive the environment reward $r_{v \cdot \tau_{\mathrm{volley}}+j}$
10:            $r'_v \leftarrow r'_v + r_{v \cdot \tau_{\mathrm{volley}}+j}$
11:        **end if**
12:    **end for**
13:    $v \leftarrow v + 1$
14: **end while**

---

**Algorithm 3** DDPG-based Roll-out for Low-level Action Policy Training

---

1: **Input:** high-level DLM policy $\pi(\cdot|\theta_{\mathrm{DLM}})$ as described in Eq. (2), low-level action policies $\{\pi_i(\cdot|\theta_i)\}$, low-level critic $\{Q_i(\cdot|\theta_{Q_i})\}$, horizon $H$, volley size $\tau_{\mathrm{volley}}$, learning rate for actor $\alpha$, learning rate for critic $\beta$
2: $t \leftarrow 0$, observe the environment state $s_0$
3: **while** task not completed and $t < H$ **do**
4:     calculate input predicates $\mathcal{P}$ based on $s_t$
5:     sample an index $\delta_t \sim \pi(s_t|\theta_{\mathrm{DLM}})$
6:     **for** $j \leftarrow 1$ **to** $\tau_{\mathrm{volley}}$ **do**
7:         obtain action $a_{t+j}$ from $\pi_{\delta_t}(s_{t+j}|\theta_{\delta_t})$
8:         receive the reward $r_{t+j}$, observe the new environment state $s_{t+j+1}$
9:         get estimated value $\omega_{t+j}$ from the critic network of DLM , get $\delta'$ from DLM
10:        $\theta_{\delta_t} \leftarrow \nabla(\log \pi_{\theta_{\delta_t}}(a'_v|s'_v)Q_{\delta_t}(s_{t+j-1}, a_{t+j-1})) + \alpha \cdot \theta_{\delta_t}$
11:        $\theta_{Q_{\delta_t}} \leftarrow \nabla(R + \gamma Q_{\delta_t}(s_{t+j}, a_{t+j}) - Q_{\delta_t}(s_{t+j-1}, a_{t+j-1})) + \beta \cdot \theta_{Q_{\delta_t}}$
12:    **end for**
13:    $t \leftarrow t + \tau_{\mathrm{volley}}$
14: **end while**

---

# E    AUTOMATON REPRESENTATION FOR HIGHWAY TASK

The automaton before simplification using the Hopcroft algorithm is presented in Figure 3. When we compare this with the reduced version in Figure 4(a), several state reductions can be observed:

• The final states $q_6, q_9, q_{14}, q_{17}, q_{21}, q_{24}$ are grouped into $q_{29}$.

• $q_{10}, q_{11}$ and $q_{18}$ are grouped into $q_{30}$.

• $q_2, q_{12}$ and $q_{19}$ are grouped into $q_{25}$.

• $q_7, q_{15}$ and $q_{22}$ are grouped into $q_{26}$.

• $q_4, q_{13}$ and $q_{20}$ are grouped into $q_{27}$.

• $q_8, q_{16}$ and $q_{23}$ are grouped into $q_{28}$.

We also detail the predicate representation for states $q_i$. The predicate representations of the automaton states can be seen in Eq. 13. The agent is regarded as in state $q_i$ when the logical expression for $q_i$ holds.

$$
\begin{aligned}
q_1 &\leftarrow P_1 \wedge \neg P_2 \wedge \neg P_3 \wedge \neg P_4 \\
q_{25} &\leftarrow \neg P_1 \wedge \neg P_2 \wedge \neg P_3 \wedge P_4 \\
q_{26} &\leftarrow \neg P_1, \wedge \neg P_2 \wedge P_3 \wedge \neg P_4 \\
q_{27} &\leftarrow \neg P_1 \wedge \neg P_2 \wedge \neg P_3 \wedge \neg P_4 \wedge P_{85} \\
q_{28} &\leftarrow \neg P_1 \wedge \neg P_2 \wedge \neg P_3 \wedge \neg P_4 \wedge P_{87} \\
q_{29} &\leftarrow \neg P_1 \wedge \neg P_3 \wedge \neg P_4 \wedge P_{86} \\
q_{30} &\leftarrow P_1 \wedge \neg P_2 \wedge \neg P_3 \wedge \neg P_4 \wedge (P_8 \vee P_9)
\end{aligned}
\tag{13}
$$

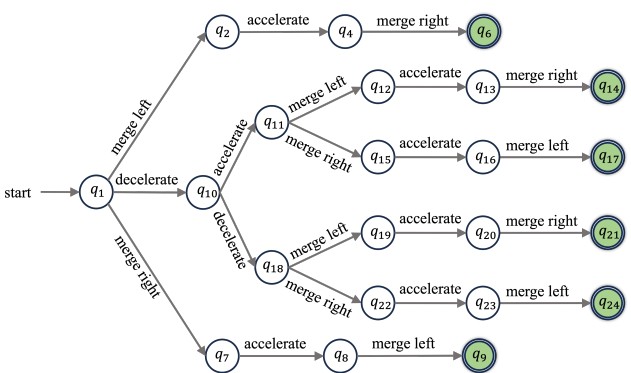

Figure 3: **Automaton before reduced for *Highway* environment.**

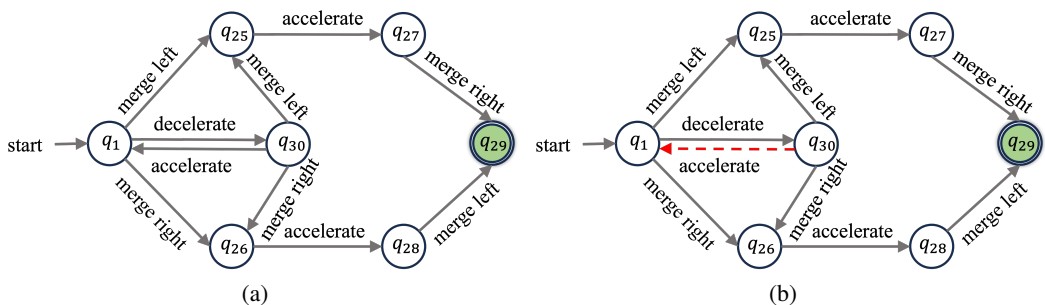

(a)          (b)

Figure 4: **Automaton representation and refinement in *Highway*.** (a) The reduced automaton for *Highway*. (b) The automaton after the expert refinement. The edge of the automaton represents the low-level action policy.

Here we list the key predicates mentioned in Eq. 13, and they are all contained in Table 7.

$P_1$:     if there is a car ahead.
$P_2$:     if there is a car behind.
$P_3$:     if there is a car on the left.
$P_4$:     if there is a car on the right.
$P_{85}$:    if the ego car is on the left of the target lane.
$P_{86}$:    if the ego car is on the target lane.
$P_{87}$:    if the ego car is on the right of the target lane.
$P_8$:     if the x-axis relative distance of the car ahead and ego car is between 5 and 10.
$P_9$:     if the x-axis relative distance of the car ahead and ego car is larger than 10.

From Figure 4 and Eq. 13, we can describe each automaton state in the overtaking task. We start from $q_1$, where there is a car in front of the ego agent. Then the ego agent can take three feasible actions: *decelerate*, *merge left*, *merge right*. If the ego agent chooses to decelerate, it will reach $q_{30}$. $q_{30}$ and $q_1$ are almost the same except that the distances between the two cars become larger. If it takes left (right) lane change action, we can find the value-focused predicate $P_4(P_3)$ changes. Then the agent accelerates until key predicates $P_4, P_3$ both become false, which means it is a proper time to get to the origin lane. Finally, it takes a right (left) lane change to finish overtaking (reach $q_{29}$).

The automaton after *expert refinement* is shown in Figure 4(b). The edge from $q_{30} \rightarrow q_1$ is eliminated through extra expert knowledge $P_{expert} = last\ action\ is\ deceleration \wedge P_1$ to prevent the car agent from repeating *accelerate* and *decelerate*. Since we include this more aggressive expert predicate, we can see a significant increase in the average velocity in Table 3.

## F    AUTOMATON REPRESENTATION FOR FETCH TASK

Here we present more detailed information about the automaton generation and expert refinement for *PickLiftPlace* Task. The automaton of *PickLiftPlace* task before reduction is presented in Figure 5, and several state reductions can be observed:

• The final states $q_5, q_6$ are grouped into $q_8$.

• States $q_3, q_4$ are grouped into $q_7$.

Figure 6(b) is the automaton after the expert refinement. It is decomposed into 5 states. $q_{10}$ is a final state, which represents that the whole *Fetch* task succeeds. The edges represent low-level action policies. These policies can be concluded as approach, grab, lift, and reach the target position. At $q_7$, we have 2 paths that can lead to the final state. This is because when the target state is above the horizon, lift and reach can be further combined into one policy, which is the shortcut edge from $q_7$ to $q_{10}$. From this perspective, ReLIC can also generate its high-level policy instead of executing low-level policy in a sequential arrangement.

We add additional expert knowledge $P_0^e = |x_{object} - x_{gripper}| + |y_{object} - y_{gripper}| + |z_{object} - z_{gripper}| + |g_{left} + g_{right} - t| \leq err, P_1^e = |y_{object} - \delta_z| \leq err$, where $z$ represents the z-axis position of the object, $t$ represents simulation time, and $\delta_z$ represents the threshold height set for the current task, $err$ is a tolerable error range, to the high-level model and fine-tune it to get the refined automaton (shown in Figure 6(b)).

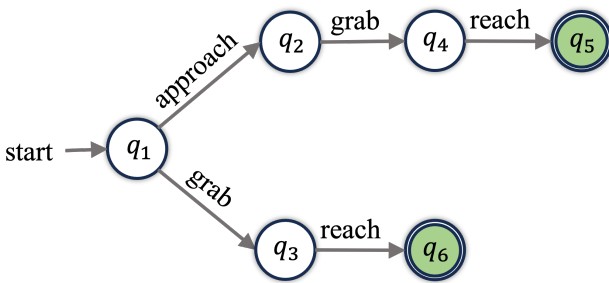

Figure 5: **Automaton before reduced for *Fetch* environment.**

$$q_1 \leftarrow P_{56} \wedge P_{57} \wedge P_{58} \wedge P_{59} \wedge P_{60} \wedge P_{61} \wedge P_{63}$$
$$q_2 \leftarrow P_{21} \wedge P_{17} \wedge P_{18} \wedge P_{59} \wedge P_{60} \wedge P_{61} \wedge P_{63}$$
$$q_7 \leftarrow P_{21} \wedge P_{17} \wedge P_{18} \wedge P_9 \wedge P_{10} \wedge P_{61} \wedge P_{63} \tag{14}$$
$$q_9 \leftarrow P_{21} \wedge P_{17} \wedge P_{18} \wedge P_9 \wedge P_{10} \wedge P_{62} \wedge P_{63} \wedge P_1^e$$
$$q_{10} \leftarrow P_{21} \wedge P_{17} \wedge P_{18} \wedge P_9 \wedge P_{10} \wedge P_{62} \wedge P_{64} \wedge P_0^e$$

Here we list the key predicates mentioned in Eq. 14, and they are all contained in Table 8.

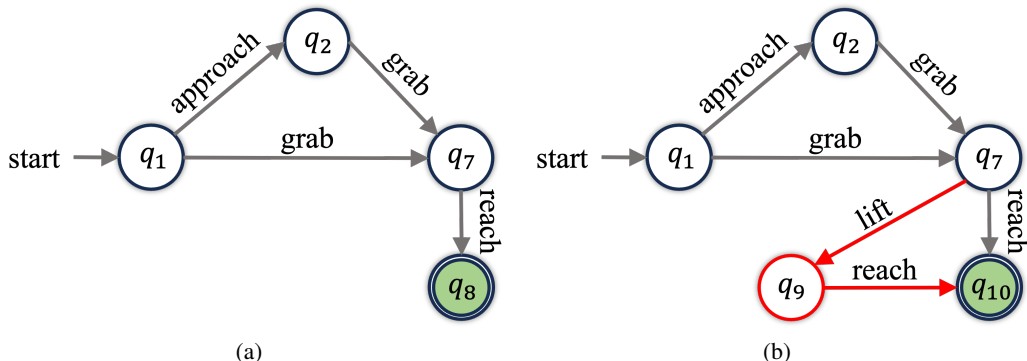

Figure 6: **Automaton Representation and refinement in *Fetch*.** (a) The reduced automaton for *Fetch*. (b) The automaton after the expert refinement. Every edge of the automaton represents the low-level action policy.

| | |
|---|---|
| $P_{61}$: | the height of the object is lower than the target height 0.45 |
| $P_{62}$: | the height of the object is lower than the target height 0.45 |
| $P_{63}$: | the object has not reached the target point |
| $P_{64}$: | the object has reached the target point |
| $P_{56}$: | the x-axis relative distance of the gripper and the object is larger than 0.1 |
| $P_{21}$: | the x-axis relative distance of the gripper and the object is between 0.008 and 0.01 |
| $P_{57}$: | the y-axis relative distance of the gripper and the object is larger than 0.1 |
| $P_{17}$: | the y-axis relative distance of the gripper and the object is between 0.006 and 0.008 |
| $P_{58}$: | the z-axis relative distance of the gripper and the object is larger than 0.1 |
| $P_{18}$: | the z-axis relative distance of the gripper and the object is between 0.006 and 0.008 |
| $P_{59}$: | the displacement of the left claw is larger than 0.1 |
| $P_9$ : | the displacement of the left claw is between 0.002 and 0.004 |
| $P_{60}$: | the displacement of the right claw is larger than 0.1 |
| $P_{10}$: | the displacement of the right claw is between 0.002 and 0.004 |

By abstracting our high-level policy into an automaton and extracting the corresponding predicates for each key node, we show the capability of our logical model to learn more complex logic beyond sequential logic, and the effectiveness and uniqueness of our predicate descriptions of the states.

## G   DFA Synthesize Algorithm

Here we provide a detailed description of our automaton synthesis program. The output of the high-level logical model is a probability distribution for each low-level action policy. We select the action policy $\pi_\delta$ based on this probability distribution. We invoke the corresponding policy $\pi_\delta$ also for many consecutive periods. During this process, we track the value of the *key predicate*. We define all the observations $s$ with the same *key predicate* as a new state $q$ for the automaton. We group the state $s_{auto_v}$ with the same predecessor automaton state $s_{auto_{v-1}}$ and transition $\delta_{v-1}$ into a collection, which is represented by a new state $S_i$ in the automaton. This process is repeated until the end of the episode.

In the experiment, we collect a large amount of traces (specifically 100,000) and group the observations into different automaton states. Additionally, we employ a predicate for judging whether the task is accomplished, so that we can easily figure out the terminating state for the automaton. Finally, we apply the Hopcroft Algorithm to simplify the automaton.

The pseudo-code for this algorithm is Algorithm 4.

**Algorithm 4** Synthesis Automaton Logic Representation for High-level Policy

---

**Input:** high-level DLM policy $\pi(\cdot|\theta_{\mathrm{DLM}})$ as described in Eq. (2), low-level action policies $\{\pi_i(\cdot|\theta_i)\}$, horizon $H$, volley size $\tau_{\mathrm{volley}}$, epoch $N$.

epoch count $n \leftarrow 0$

automaton state node map $\mathcal{M}$, the key of $\mathcal{M}$ is the state node of automaton, while the value is another submap describing the decision and the corresponding next state

**while** $n < N$ **do**

    Volley count $v \leftarrow 0$

    **while** task not completed and $v < H/\tau_{\mathrm{volley}}$ **do**

        calculate the input predicates $\mathcal{P}$ based on $s_v$

        sample an index $\delta_v \sim \pi(s_v|\theta_{\mathrm{DLM}})$

        **if** $\delta_v \neq \delta_{v-1}$ **then**

            extract key predicates $\mathcal{P}^*$ from $\pi(\cdot|\theta_{\mathrm{DLM}})$ for the output $\delta_v$

            calculate the true value of $\mathcal{P}^*$ based on $s_v$

            automaton state $S_{auto_v} \leftarrow \mathcal{P}^*$

            **if** $S_{auto_{v-1}}$ in $\mathcal{M}$ **then**

                **if** ($\delta_{v-1}$ in $\mathcal{M}[S_{auto_{v-1}}]$ **then**

                    $S_{exist} \leftarrow \mathcal{M}[S_{auto_{v-1}}][\delta_{v-1}]$

                    merge $S_{exist}$ and $S_{auto_v}$ because they represent the same state in automaton;

                **else**

                    add $\{\delta_{v-1} : S_{auto_v}\}$ to $\mathcal{M}[S_{auto_{v-1}}]$

                **end if**

            **else**

                add $\{S_{auto_{v-1}} : \{(\delta_{v-1} : S_{auto_v})\}\}$ to $\mathcal{M}$

            **end if**

        **end if**

        **for** $j \leftarrow 0$ **to** $\tau_{\mathrm{volley}}$ **do**

            if $j \neq 0$ then observe the environment state $s_{v \cdot \tau_{\mathrm{volley}}+j}$

            execute the environment action $a_{v \cdot \tau_{\mathrm{volley}}+j} \leftarrow \pi_{\delta_v}(s_{v \cdot \tau_{\mathrm{volley}}+j}|\theta_{\delta_v})$

        **end for**

    **end while**

    $v \leftarrow v + 1$

**end while**

Split all nodes into final state $A$ and non-final state $N$

$N \leftarrow \{S \backslash FinalState\}$

**while** True **do**

    **for** each state set $\mathcal{T}$ in N **do**

        **for** each $\delta$ in option set **do**

            **if** $\delta$ can split $\mathcal{T}$ **then**

                split $\mathcal{T}$ into $\mathcal{T}_1 ... \mathcal{T}_k$

                add $\mathcal{T}_1 ... \mathcal{T}_k$ to N

            **end if**

        **end for**

    **end for**

    **if** no split operation is done **then**

        break

    **end if**

**end while**

---

Table 5: **Hyperparameters in *Highway* environment.**

| Hyperparameter | Value |
|---|---|
| Joint Training Epoch | 500 |
| expert Refinement Frequency | 50 |
| DLM Depth | 7 |
| DLM Breadth | 3 |
| DLM Discount Factor | 0.99 |
| DLM Policy Number | 4 |
| DDPG Discount Factor | 0.99 |
| DDPG Learning Rate | 0.0005 |
| DDPG Replay Buffer Size | 50000 |

Table 6: **Hyperparameters in *Fetch-Pick-And-Place* environment.**

| Hyperparameter | Value |
|---|---|
| Joint Training Epoch | 500 |
| expert Refinement Frequency | 50 |
| DLM Depth | 3 |
| DLM Breadth | 3 |
| DLM Discount Factor | 0.99 |
| DLM Policy Number | 4 |
| DDPG Discount Factor | 0.95 |
| DDPG Learning Rate | 0.0001 |
| DDPG Replay Buffer Size | 200000 |

## H IMPLEMENTATION DETAILS

All experiments were conducted on a machine running Ubuntu 22, equipped with an Intel Xeon 2.5 GHz CPU, 32 GB RAM, and an NVIDIA A100 GPU.

### H.1 HIGHWAY ENVIRONMENT

In the *Highway* environment, we have 4 low-level action policies corresponding to `acceleration`, `deceleration`, `merge left`, `merge right`. We choose the Deep Deterministic Policy Gradient (DDPG) algorithm for low-level action policies. We use Adam optimizer to update the parameters in the DDPG model.

The Hyperparameters for the *Highway* environment are shown in Table 5.

### H.2 FETCH ENVIRONMENT

In the *Fetch* environment, we conduct three experiments *Pick&Place*, *Pick&PlaceCorner*, and *Pick-LiftPlace*. We have 4 low-level action policies corresponding to `approach`, `grab`, `lift`, `reach`. We choose the DDPG algorithm for low-level action policies. We use Adam optimizer to update the parameters in the DDPG model.

The Hyperparameters for the *Fetch* environment are shown in Table 6.

## I PREDICATES SUMMARY

In this section, we provide a summary of all input predicates and their corresponding relationship with the input states for our two experiments: *Highway* and *Fetch-Pick-And-Place*.

### I.1 INPUT PREDICATES IN HIGHWAY ENVIRONMENT

Here we show the mathematical form of input predicates, which is derived from input states in *Highway* Environment. The specific input states and predicates are listed in Table 7.

The meanings of the variables in the input states are as follows:

$d_{x0}$: the x-axis position of the ego agent.

$d_{x1}$: the x-axis position of the nearest car ahead.

$d_{x2}$: the x-axis position of the nearest car behind.

$d_{x3}$: the x-axis position of the nearest car on the left.

$d_{x4}$: the x-axis position of the nearest car on the right.

$d_{y0}$: the y-axis position of the ego agent.

$d_{y1}$: the y-axis position of the nearest car ahead.

$d_{y2}$: the y-axis position of the nearest car behind.

$d_{y2}$: the y-axis position of the nearest car behind.

$d_{y3}$: the y-axis position of the nearest car on the left.

$d_{y4}$: the y-axis position of the nearest car on the right.

$v_{x0}$: the x-axis velocity of the ego agent.

$v_{x1}$: the x-axis velocity of the car ahead.

$v_{x2}$: the x-axis velocity of the car behind.

$v_{x3}$: the x-axis velocity of the car on the left.

$v_{x4}$: the x-axis velocity of the car on the right.

$v_{y0}$: the y-axis velocity of the ego agent.

$v_{y1}$: the x-axis velocity of the car ahead.

$v_{y2}$: the x-axis velocity of the car behind.

$v_{y3}$: the x-axis velocity of the car on the left.

$v_{y4}$: the x-axis velocity of the car on the right.

$e_0$: if there exists a car ahead.

$e_1$: if there exists a car behind.

$e_2$: if there exists a car on the left.

$e_3$: if there exists a car on the right.

$l_0$: the lane in which the ego agent is located.

$l_1$: the target lane.

## I.2 INPUT PREDICATES IN FETCH-PICK-AND-PLACE ENVIRONMENT

Here we show the mathematical form of input predicates, which is derived from input states in *Fetch-Pick-And-Place* environment.

We set the activating intervals as follows: {0, 0.002, 0.004, 0.006, 0.008, 0.01, 0.012, 0.014, 0.016, 0.018, 0.02, 0.026, 1}. They are used to divide the input states into discrete predicates as the input of the high-level logical model. The specific input states and predicates are listed in Table 8. Except for those predicates, $P_{61}$ and $P_{62}$ represent if the height of the object is higher than the target height or not based on $z_1$, while $P_{63}$, $P_{64}$ represent if the object has reached the target position or not based on $(x_1, y_1, z_1)$.

The meanings of the variables in the Input States are as follows:

$x_0$: The x-axis position of the gripper.

$x_1$: The x-axis position of the object.

$y_0$: The y-axis position of the gripper.

$y_1$: The y-axis position of the object.

$z_0$: The z-axis position of the gripper.

$z_1$: The z-axis position of the object.

$d_0$: The displacement of the left claw.

$d_1$: The displacement of the right claw.

## J PROMPT TEMPLATE OVERVIEW

We present the complete prompt template for expert refinement via LLM (§ 3.3.2). The prompt is generally composed of several parts:

- Task description: Provide LLM with a description of the task and background information of the environment.

- Input description: Describe the semantic meaning of each input variable in the form of an interpretable sentence and the relation between input variables.

- Input: List all input variables in the sequence of the input description.

- Your Task: Input the reduced automaton and failure traces.

- Output: Instruct the LLM to generate the formatted result by employing the chain-of-thought method.

Table 9 demonstrates the complete prompt template of the *Fetch* task. Table 10 demonstrates the complete prompt template of the *Highway* task. Table 11 presents an example of the input format.

## K THE USE OF LARGE LANGUAGE MODELS

In the process of drafting this paper, we employed large language models (LLMs) as an auxiliary tool to enhance the quality and clarity of our written English. The primary application was to identify and correct grammatical inaccuracies, refine sentence structures, and polish academic expressions, thereby improving the overall readability and professionalism of the manuscript.

Table 7: **Input predicates in *Highway* environment.**

| Activating Intervals | Input States | Predicates | Description |
|---|---|---|---|
| $\{0, 1, 2.5, 5, 10, \infty\}$ | $|d_{x0} - d_{x1}|$ | $P_5, P_6, P_7, P_8, P_9$ | The x-axis relative distance between the ego agent and the car ahead. |
| $\{0, 1, 2.5, 5, 10, \infty\}$ | $|d_{x0} - d_{x2}|$ | $P_{10}, P_{11}, P_{12}, P_{13}, P_{14}$ | The x-axis relative distance between the ego agent and the car behind. |
| $\{0, 1, 2.5, 5, 10, \infty\}$ | $|d_{x0} - d_{x3}|$ | $P_{15}, P_{16}, P_{17}, P_{18}, P_{19}$ | The x-axis relative distance between the ego agent and the car on the left. |
| $\{0, 1, 2.5, 5, 10, \infty\}$ | $|d_{x0} - d_{x4}|$ | $P_{20}, P_{21}, P_{22}, P_{23}, P_{24}$ | The x-axis relative distance between the ego agent and the car on the right. |
| $\{0, 1, 2.5, 5, 10, \infty\}$ | $|d_{y0} - d_{y1}|$ | $P_{25}, P_{26}, P_{27}, P_{28}, P_{29}$ | The y-axis relative distance between the ego agent and the car ahead. |
| $\{0, 1, 2.5, 5, 10, \infty\}$ | $|d_{y0} - d_{y2}|$ | $P_{30}, P_{31}, P_{32}, P_{33}, P_{34}$ | The y-axis relative distance between the ego agent and the car behind. |
| $\{0, 1, 2.5, 5, 10, \infty\}$ | $|d_{y0} - d_{y3}|$ | $P_{35}, P_{36}, P_{37}, P_{38}, P_{39}$ | The y-axis relative distance between the ego agent and the car on the left. |
| $\{0, 1, 2.5, 5, 10, \infty\}$ | $|d_{y0} - d_{y4}|$ | $P_{40}, P_{41}, P_{42}, P_{43}, P_{44}$ | The y-axis relative distance between the ego agent and the car on the right. |
| $\{0, 0.5, 1, 3, 6, \infty\}$ | $|v_{x0} - d_{x1}|$ | $P_{45}, P_{46}, P_{47}, P_{48}, P_{49}$ | The x-axis relative velocity between the ego agent and the car ahead. |
| $\{0, 0.5, 1, 3, 6, \infty\}$ | $|v_{x0} - d_{x2}|$ | $P_{50}, P_{51}, P_{52}, P_{53}, P_{54}$ | The x-axis relative velocity between the ego agent and the car behind. |
| $\{0, 0.5, 1, 3, 6, \infty\}$ | $|v_{x0} - d_{x3}|$ | $P_{55}, P_{56}, P_{57}, P_{58}, P_{59}$ | The x-axis relative velocity between the ego agent and the car on the left. |
| $\{0, 0.5, 1, 3, 6, \infty\}$ | $|v_{x0} - d_{x4}|$ | $P_{60}, P_{61}, P_{62}, P_{63}, P_{64}$ | The x-axis relative velocity between the ego agent and the car on the right. |
| $\{0, 0.5, 1, 3, 6, \infty\}$ | $|v_{y0} - d_{y1}|$ | $P_{65}, P_{66}, P_{67}, P_{68}, P_{69}$ | The y-axis relative velocity between the ego agent and the car ahead. |
| $\{0, 0.5, 1, 3, 6, \infty\}$ | $|v_{y0} - d_{y2}|$ | $P_{70}, P_{71}, P_{72}, P_{73}, P_{74}$ | The y-axis relative velocity between the ego agent and the car behind. |
| $\{0, 0.5, 1, 3, 6, \infty\}$ | $|v_{y0} - d_{y3}|$ | $P_{75}, P_{76}, P_{77}, P_{78}, P_{79}$ | The y-axis relative velocity between the ego agent and the car on the left. |
| $\{0, 0.5, 1, 3, 6, \infty\}$ | $|v_{y0} - d_{y4}|$ | $P_{80}, P_{81}, P_{82}, P_{83}, P_{84}$ | The y-axis relative velocity between the ego agent and the car on the right. |
| $\{\}$ | $e_i == 1, \ i = \{0, 1, 2, 3\}$ | $P_1, P_2, P_3, P_4$ | If there exists a car ahead / behind / on the left / on the right. |
| $\{-\infty, -0.1, 0.1, \infty\}$ | $l_0 - l_1$ | 21 
 $P_{85}, P_{86}, P_{87}$ | The relative direction between the lane in which the ego agent is located and the target lane. |

Table 8: **Input predicates in *Fetch-Pick-And-Place* environment.**

| Input States | Predicates | Description |
|---|---|---|
| $|x_0 - x_1|$ | $P_1, P_6, P_{11}, P_{16}, P_{21}, P_{26}, P_{31},$ $P_{36}, P_{41}, P_{46}, P_{51}, P_{56}$ | The x-axis relative distance between the gripper and the object. |
| $|y_0 - y_1|$ | $P_2, P_7, P_{12}, P_{17}, P_{22}, P_{27}, P_{32},$ $P_{37}, P_{42}, P_{47}, P_{52}, P_{57}$ | The y-axis relative distance between the gripper and the object. |
| $|z_0 - z_1|$ | $P_3, P_8, P_{13}, P_{18}, P_{23}, P_{28}, P_{33},$ $P_{38}, P_{43}, P_{48}, P_{53}, P_{58}$ | The z-axis relative distance between the gripper and the object. |
| $|d_0|$ | $P_4, P_9, P_{14}, P_{19}, P_{24}, P_{29}, P_{34},$ $P_{39}, P_{44}, P_{49}, P_{54}, P_{59}$ | The displacement of the left claw. |
| $|d_1|$ | $P_5, P_{10}, P_{15}, P_{20}, P_{25}, P_{30}, P_{35},$ $P_{40}, P_{45}, P_{50}, P_{55}, P_{60}$ | The displacement of the right claw. |

Specifically, selected paragraphs or sentences from our initial drafts were input into an LLM (e.g., DeepSeek-v3.1 or a comparable model) with explicit instructions focused solely on language checking and polishing. The prompts were designed to request grammatical corrections, suggestions for more concise or academically appropriate phrasing, and improvements in logical flow, without altering the core technical content or scientific meaning.

It is crucial to emphasize that the role of the LLM was strictly limited to that of a writing assistant. All substantive intellectual contributions, including the core ideas, theoretical framework, experimental design, data analysis, and result interpretation, remain entirely our own. The final decision to adopt any suggestion provided by the LLM was always subject to our careful review and judgment. We ensured that every change aligned with our intended meaning and adhered to the standards of academic integrity.

This use of LLMs significantly streamlined the writing and revision process, allowing us to focus more effectively on the scientific rigor and conceptual depth of our work.

Table 9: **Complete prompt template for the _Fetch_ environment.**

You are an expert in refining the automaton of a robot for the task Fetch available in the OpenAI Gym repository. You need to first understand the task and the automaton of the robot.

# Task Description

The task in the environment is for a manipulator to move a block to a target position on top of a table or in mid-air. The robot is a 7-DoF Fetch Mobile Manipulator with a two-fingered parallel gripper. The robot is controlled by small displacements of the gripper in Cartesian coordinates, and the inverse kinematics are computed internally by the MuJoCo framework. The gripper can be opened or closed in order to perform the grasping operation of pick and place. The task is also continuing, which means that the robot has to maintain the block in the target position for an indefinite period.

# Total Predicates ID and Its Description

- total_predicate is a dictionary that maps the predicate ID to its description, including three types of predicates:
1. The relative distance between the end effector and the object along the x, y, and z axes.
2. The displacement of the left gripper and the right gripper.
3. The height of the object from the table.

# Key Predicates ID

- *key_predicates_id* is a list of predicate IDs that are important for the task.

# State Predicates

- *state_predicates* is a dictionary that maps the state ID in *states* to a list of tuples of predicates.
- The boolean values of each predicate tuple are in the order of the *key_predicates_id*. # Automaton
- *action* is a dictionary that maps the action letter to its description.
- *states* is a list of state IDs in the automaton.
- *start_state* is the initial state of the automaton.
- *accept_states* is a list of accepting states in the automaton.
- *transitions* is a dictionary of transitions in the automaton. The key is a tuple of the state ID and the action letter, and the value is the next state ID.

# Failure Trajectory

- *failure_trajectory* is a list of episodes. Each episode is a dictionary of transitions. The key is a tuple of the state, and the value is the next state. The state is a tuple represented by boolean values in the order of the *key_predicates_id and the action letter.

# Input

## Total Predicates ID and Its Description
- total_predicates = {TOTAL_PREDICATES}

## Key Predicates ID
- key_predicates_id = {KEY_PREDICATES_ID}

## State Predicates
- state_predicates = {STATE_PREDICATES}

## Automaton
- action = {ACTION}
- states = {STATES}
- start_state = {START_STATES}
- accept_states = {ACCEPT_STATES}
- transitions = {TRANSITIONS}

## Failure Trajectory
- failure_trajectory = {FAILURE_TRAJECTORY}

# Your Task

You need to analyze and refine the automaton of the robot. You must follow the following rules.
1. You can also leverage your own knowledge about the goal of the task, but the conclusions must be based on the Input.
2. You need to analyze the automaton in these three steps: (a) analyze the logical relation between key predicates of states, (b) analyze why failed trajectories failed to reach accepting states, and (c) refine the automaton by proposing new key predicates.
3. When performing (a), you can first consider the relationship between predicate tuples in the list and then consider using the logical operators AND, OR, and NOT to combine the predicates in the tuple.
4. When performing (c), you should reduce the number of states to four by removing counterintuitive transitions.
5. When performing (c), the format of the new states and transitions must be consistent with the existing automaton.
6. The state ID must be a unique integer, and the action letter must be a unique character.

## Output

Now, analyze the logical relation between key predicates of states.
{ChatGPT response}
Analyze why failed trajectories failed to reach accepted states.
{ChatGPT response}
Refine the automaton by proposing new key predicates.
{ChatGPT response}

Table 10: **Complete prompt template for the *Highway* environment.**

You are an expert in refining the automaton of an ego vehicle for the task Highway available in the OpenAI Gym repository. You need to first understand the task and the automaton of the ego vehicle.

# Task Description

The task in the environment is to drive an ego vehicle as fast as possible. At the same time, the ego vehicle should not hit any other cars. The vehicle is controlled by linear acceleration and angular acceleration in Cartesian coordinates, and the inverse kinematics are computed internally by the Highway environment. There are four lanes in the same direction in the environment. Other vehicles in the environment travel at a certain speed on a specific lane and will not perform unconventional driving maneuvers. This task lasts for a fixed duration, during which the vehicle must keep moving continuously.

# Total Predicates ID and Its Description

- total_predicate is a dictionary that maps the predicate ID to its description, including three types of predicates:
1. The relative distance between the ego vehicle and the car along the x and y axes.
2. The relative velocity between the ego vehicle and the car along the x-axis.
3. If there exists a car ahead, behind, on the left, or the right.
4. The relative direction between the lane in which the ego agent is located and the target lane.

# Key Predicates ID

- *key_predicates_id* is a list of predicate IDs that are important for the task.

# State Predicates

- *state_predicates* is a dictionary that maps the state ID in *states* to a list of tuples of predicates.
- The boolean values of each predicate tuple are in the order of the *key_predicates_id*. # Automaton
- *action* is a dictionary that maps the action letter to its description.
- *states* is a list of state IDs in the automaton.
- *start_state* is the initial state of the automaton.
- *accept_states* is a list of accepting states in the automaton.
- *transitions* is a dictionary of transitions in the automaton. The key is a tuple of the state ID and the action letter, and the value is the next state ID.

# Failure Trajectory

- *failure_trajectory* is a list of episodes. Each episode is a dictionary of transitions. The key is a tuple of the state, and the value is the next state. The state is a tuple represented by boolean values in the order of the *key_predicates_id and the action letter.

# Input

## Total Predicates ID and Its Description
- total_predicates = {TOTAL_PREDICATES}
## Key Predicates ID
- key_predicates_id = {KEY_PREDICATES_ID}
## State Predicates
- state_predicates = {STATE_PREDICATES}
## Automaton
- action = {ACTION}
- states = {STATES}
- start_state = {START_STATES}
- accept_states = {ACCEPT_STATES}
- transitions = {TRANSITIONS}
## Failure Trajectory
- failure_trajectory = {FAILURE_TRAJECTORY}
# Your Task

You need to analyze and refine the automaton of the ego vehicle. You must follow the following rules.
1. You can also leverage your own knowledge about the goal of the task, but the conclusions must be based on the Input.
2. You need to analyze the automaton in these three steps: (a) analyze the logical relation between key predicates of states, (b) analyze why failed trajectories failed to reach accepting states, and (c) refine the automaton by proposing new key predicates.
3. When performing (a), you can first consider the relationship between predicate tuples in the list and then consider using the logical operators AND, OR, and NOT to combine the predicates in the tuple.
4. When performing (c), you should reduce the number of states to four by removing counterintuitive transitions.
5. When performing (c), the format of the new states and transitions must be consistent with the existing automaton.
6. The state ID must be a unique integer, and the action letter must be a unique character.
## Output

Now, analyze the logical relation between key predicates of states.
{ChatGPT response}
Analyze why failed trajectories failed to reach accepted states.
{ChatGPT response}
Refine the automaton by proposing new key predicates.
{ChatGPT response}

Table 11: **An example of variables that require the user to input.**

# Input
## Total Predicates ID and Its Description
- total_predicates = {'P0' : 'The x-axis relative distance of the end effector and the object is between 0 and 0.002.', ..., 'P55' : 'The x-axis relative distance of the end effector and the object is between 0.026 and 1.', 'P1' : 'The y-axis relative distance of the end effector and the object is between 0 and 0.002.', ..., 'P56' : 'The y-axis relative distance of the end effector and the object is between 0.026 and 1.', 'P2' : 'The z-axis relative distance of the end effector and the object is between 0 and 0.002.', ..., 'P57' : 'The z-axis relative distance of the end effector and the object is between 0.026 and 1.', 'P3' : 'The displacement of the left gripper is between 0 and 0.002.', ..., 'P58' : 'The displacement of the left gripper is between 0.026 and 1.', 'P4' : 'The displacement of the right gripper is between 0 and 0.002.', ..., P59' : 'The displacement of the right gripper is between 0.026 and 1.', 'P60' : 'The height of the object is lower than the target height 0.45.', 'P61' : 'The height of the object is higher than the target height 0.45.', }
## Key Predicates ID
- key_predicates_id = ['P35', 'P36', 'P5', 'P40', 'P41', 'P10', 'P16', 'P51', 'P56', 'P57', 'P58', 'P31', ]
## State Predicates
- state_predicates = {0 : [(0, 0, 0, 0, 0, 0, 0, 1, 0, 0, 1, 0), (0, 0, 1, 0, 1, 0, 0, 0, 1, 1, 0), (0, 0, 1, 0, 0, 0, 0, 0, 1, 0, 1, 0), (0, 0, 0, 0, 0, 0, 0, 0, 0, 1, 1, 1), (0, 0, 0, 0, 0, 1, 0, 0, 1, 0, 1, 0), (0, 0, 0, 0, 0, 0, 0, 0, 0, 0, 1, 0), (1, 0, 0, 0, 0, 0, 0, 1, 0, 1, 0), (0, 0, 0, 1, 0, 0, 0, 0, 1, 1, 0), (0, 0, 1, 0, 0, 0, 1, 0, 0, 1, 1, 0), (0, 0, 0, 0, 0, 1, 0, 1, 0, 0, 1, 0), (0, 0, 1, 0, 0, 0, 0, 0, 1, 1, 0), (0, 0, 0, 0, 0, 0, 0, 1, 0, 1, 1, 0), (0, 0, 0, 0, 0, 0, 1, 0, 0, 1, 1, 0), (0, 0, 1, 0, 0, 0, 0, 1, 0, 1, 1, 0), (1, 0, 0, 0, 0, 0, 0, 1, 1, 0), (0, 0, 0, 0, 0, 0, 1, 0, 0, 1, 1, 0), (0, 0, 0, 0, 0, 0, 0, 0, 0, 0, 0, 1, 1), ], 1 : [(0, 0, 1, 0, 0, 0, 0, 0, 0, 1, 0), (0, 0, 1, 0, 0, 0, 0, 0, 1, 1, 1), (0, 0, 0, 1, 0, 0, 0, 0, 0, 1, 0), (0, 0, 0, 0, 0, 0, 0, 1, 1, 1, 0), (0, 0, 1, 0, 0, 0, 0, 0, 1, 0), (1, 0, 0, 0, 0, 0, 0, 1, 0, 0, 1, 0), ], 2 : [(0, 0, 0, 1, 0, 0, 1, 0, 0, 1, 1, 0), ], 3 : [(0, 0, 0, 0, 0, 0, 1, 0, 0, 0, 1, 0), (0, 0, 0, 0, 0, 0, 0, 0, 0, 1, 1, 0), (0, 0, 0, 0, 0, 1, 0, 0, 0, 0, 1, 0), (1, 0, 0, 0, 0, 0, 0, 0, 0, 0, 1, 0), ], }
## Automaton
- action = {'A': 'approach', 'B': 'grab', 'D': 'lift'}
- states = [0, 1, 2, 3]
- start_state = 1
- accept_states = 0
- transitions = (0, 'B'): 3, (0, 'D'): 3, (0, 'A'): 0, (1, 'B'): 0, (1, 'A'): 3, (2, 'A'): 1, (3, 'B'): 3
## Failure Trajectory
- failure_trajectory = [{((0, 0, 0, 0, 0, 0, 0, 0, 1, 1, 1, 0), 'A'): (0, 0, 0, 0, 0, 1, 0, 0, 0, 0, 1, 0), ((0, 0, 0, 0, 0, 1, 0, 0, 0, 0, 1, 0), 'B'): (0, 0, 0, 0, 0, 0, 0, 0, 0, 0, 1, 1, 0), ((0, 0, 0, 0, 0, 0, 0, 0, 0, 1, 1, 0), 'D'): (0, 0, 0, 0, 0, 0, 0, 0, 0, 1, 1, 0)}, {((0, 0, 0, 0, 0, 0, 0, 0, 1, 1, 1, 0), 'A'): (0, 0, 0, 0, 0, 0, 0, 0, 0, 0, 1, 0), ((0, 0, 0, 0, 0, 0, 0, 0, 0, 0, 1, 0), 'B'): (0, 0, 0, 0, 0, 0, 0, 0, 0, 1, 1, 0), ((0, 0, 0, 0, 0, 0, 0, 0, 0, 1, 1, 0), 'D'): (0, 0, 0, 0, 0, 0, 0, 0, 0, 1, 1, 0), ((0, 0, 0, 1, 0, 0, 0, 0, 0, 1, 1, 0), 'A'): (0, 0, 0, 0, 0, 0, 0, 0, 0, 0, 1, 1), ((0, 0, 0, 0, 0, 0, 0, 0, 0, 0, 1, 1), 'B'): (0, 0, 1, 0, 0, 0, 0, 0, 0, 1, 1, 0), ((0, 0, 1, 0, 0, 0, 0, 0, 0, 1, 1, 0), 'D'): (0, 0, 0, 0, 0, 0, 0, 0, 0, 1, 1, 0)}, {((0, 0, 0, 0, 0, 0, 0, 0, 0, 1, 1, 1, 0), 'A'): (0, 0, 0, 0, 0, 1, 1, 0, 0, 0, 1, 0), ((0, 0, 0, 0, 0, 1, 1, 0, 0, 0, 1, 0), 'B'): (0, 0, 0, 0, 0, 0, 0, 0, 0, 0, 1, 1, 0), ((0, 0, 0, 0, 0, 0, 0, 0, 0, 0, 1, 1, 0), 'D'): (0, 0, 0, 0, 0, 0, 0, 0, 0, 0, 1, 1, 0)}, {((0, 0, 0, 0, 0, 0, 0, 0, 0, 1, 1, 1, 0), 'A'): (0, 0, 1, 0, 0, 0, 0, 0, 0, 1, 1), ((0, 0, 1, 0, 0, 0, 0, 0, 0, 1, 1), 'B'): (0, 0, 0, 0, 0, 0, 0, 0, 0, 1, 1, 0), ((0, 0, 0, 0, 0, 0, 0, 0, 0, 1, 1, 0), 'D'): (0, 0, 0, 0, 0, 0, 0, 0, 0, 1, 1, 0)}, {((0, 0, 0, 0, 0, 0, 0, 1, 1, 1, 0), 'A'): (0, 0, 0, 0, 1, 0, 0, 0, 0, 1, 0), ((0, 0, 0, 0, 1, 0, 0, 0, 0, 1, 0), 'B'): (0, 0, 0, 0, 0, 0, 0, 0, 0, 1, 1, 0), ((0, 0, 0, 0, 0, 0, 0, 0, 0, 1, 1, 0), 'D'): (0, 0, 0, 0, 0, 1, 0, 0, 0, 1, 1, 0), ((0, 0, 0, 0, 0, 1, 0, 0, 0, 1, 1, 0), 'D'): (0, 0, 0, 0, 0, 1, 0, 0, 0, 1, 1, 0)}, ]