# OpenReview forum: "Interpretable Reinforcement Learning with Self-Abstraction and Refinement"
_ICLR.cc/2026/Conference — ICLR 2026 Conference Withdrawn Submission_

### Official Review · Reviewer_TJWf · 2025-10-22

**Soundness:** 3
**Presentation:** 3
**Contribution:** 2
**Rating:** 6
**Confidence:** 3

**Summary:**

This paper introduces ReLIC, a hierarchical reinforcement learning framework that achieves interpretability and interactivity. It uses a high-level logical planner and low-level control policies. The core contributions are a self-abstraction module that synthesizes the learned logic into an interpretable automaton (DFA), and an interactive refinement loop that uses an LLM (GPT-4o) to analyze this automaton and suggest expert predicates to fix policy flaws.

**Strengths:**

*   The framework produces an interpretable automaton that is directly used to debug and improve the agent's policy, which is a significant step beyond post-hoc explanations.
*   ReLIC demonstrates superior performance against a wide range of baselines on difficult continuous control tasks, proving its practical effectiveness.
*   The model learns the core policy end-to-end, only engaging an expert (via LLM) for targeted refinement, making it more scalable than methods requiring full upfront symbolic specification.

**Weaknesses:**

*   The framework's performance may depend heavily on the quality and completeness of the initial, predefined predicate set. The paper would benefit from a discussion on how the system scales with a large number of predicates.
*   For extremely long-horizon or complex tasks, the synthesized automaton could become too large to be interpretable by a human or LLM, potentially limiting the approach's applicability.
*   It is unclear how well the framework, particularly the specialized low-level policies, would generalize to new tasks not seen during training.

**Questions:**

How sensitive is ReLIC to the initial set of human-defined predicates? Could you discuss the trade-offs between providing a sparse vs. a rich set of initial predicates?

---

> ### Author Response · Authors · 2025-11-21
>
> We sincerely thank the reviewer for the insightful comments and constructive suggestions, which have helped us improve the paper.
>
> **Comment 1:** The framework's performance may depend heavily on the quality and completeness of the initial, predefined predicate set. The paper would benefit from a discussion on how the system scales with a large number of predicates.
>
> **Response to comment 1:** Please see our response to Comment 4.
>
>
>
> **Comment 2:** For extremely long-horizon or complex tasks, the synthesized automaton could become too large to be interpretable by a human or LLM, potentially limiting the approach's applicability.
>
> **Response to comment 2:**
>  Before presenting the automaton to a human or an LLM, we apply the classical Hopcroft minimization algorithm to reduce it. In the appendix (Fig. 3 and Fig. 4(a)), we show how Hopcroft reduction simplifies complex automata. After minimization, the resulting automata are intuitive for both humans and LLMs to understand and edit.
>
>
>
> **Comment 3:** It is unclear how well the framework, particularly the specialized low-level policies, would generalize to new tasks not seen during training.
>
> **Response to comment 3:**
>  Please refer to our PickHighPlace experiments.
>
>
>
> **Comment 4:** How sensitive is ReLIC to the initial set of human-defined predicates? Could you discuss the trade-offs between providing a sparse vs. a rich set of initial predicates?
>
> **Response to comment 4:**
> ReLIC is not sensitive to the initial set of human-defined predicates, and our default setup already includes a sufficiently large predicate set. We conducted an ablation study by randomly adding/removing subsets of predicates from the original setup, with the total number reaching around 140.
>
> As shown in Table A, LLM refinement maintains a high success rate even when 15 predicates are removed at random. This robustness is due to: (1) some initial predicates being redundant or suboptimal, and (2) the LLM leveraging failure trajectories to introduce expert predicates that compensate for missing or inadequate ones.
>
> When many new initial predicates are added, ReLIC does not degrade substantially (unlike the removal case), because key predicates are preserved while redundant ones are effectively ignored.
>
> Table A is as follows:
>
> | Tasks            | Number of initial predicates | +40 (richest) | +15      | +10      | 0        | -10      | -15      | -40 (sparsest) |
> | ---------------- | ---------------------------- | ------------- | -------- | -------- | -------- | -------- | -------- | -------------- |
> | Pick&Place       | Success rate (%)             | 97.2±1.1      | 97.2±0.9 | 97.4±0.9 | 97.3±0.9 | 97.2±1.0 | 94.8±1.1 | 61.1±1.0       |
> | Pick&PlaceCorner | Success rate (%)             | 99.2±0.8      | 99.5±0.8 | 99.1±0.8 | 99.3±0.8 | 99.0±0.9 | 95.9±1.0 | 65.2±1.3       |
> | PickLiftPlace    | Success rate (%)             | 99.0±1.1      | 99.1±0.7 | 99.0±0.8 | 99.0±0.9 | 98.8±0.8 | 94.5±0.8 | 62.1±1.5       |
> | PickHighPlace    | Success rate (%)             | 90.5±0.9      | 91.0±0.8 | 90.5±0.7 | 90.5±1.1 | 90.0±0.8 | 86.5±1.0 | 50.3±1.2       |
> | Highway          | Length                       | 97.1±0.1      | 97.1±0.1 | 96.2±0.1 | 97.1±0.1 | 96.3±0.2 | 96.0±0.5 | 52.2±0.3       |
> | Highway          | Velocity                     | 27.9±0.1      | 27.9±0.1 | 28.3±0.1 | 27.9±0.1 | 27.9±0.1 | 27.4±0.1 | 25.3±0.2       |
> | Highway          | Crash rate                   | 4.0±0.8       | 4.2±0.7  | 4.4±0.8  | 4.0±0.8  | 5.1±0.6  | 6.2±0.7  | 23.7±1.3       |

---

> > ### Comment · Reviewer_TJWf · 2025-11-22
> >
> > Thanks for your reply. I have no further questions and will keep my score.

---

### Official Review · Reviewer_HB7F · 2025-10-29

**Soundness:** 2
**Presentation:** 3
**Contribution:** 2
**Rating:** 4
**Confidence:** 2

**Summary:**

The paper proposes **ReLIC**, a hierarchical RL method for composite tasks. ReLIC has (i) a high-level **Differentiable Logic Machine (DLM)** that reasons over predicates, (ii) **low-level continuous control policies**, and (iii) a **self-abstraction and refinement** pipeline. From training rollouts, ReLIC extracts key predicates and synthesizes a **deterministic finite automaton (DFA)** to expose high-level decisions; an **LLM (GPT-4o)** then inspects the reduced automaton plus failure trajectories, proposes new expert predicates, and these are fed back via **joint training** of the logic planner and action policies.

**Strengths:**

* **Clear objective.** Integrates interpretability (automaton from key predicates) with interactivity (LLM-driven knowledge injection) without pre-defining a logical schema.
* **Technical design.** Presents joint training of the logical planner and low-level policies, plus a pipeline to synthesize and reduce a DFA from learned behavior.
* **Experiments.** Evaluated on Highway and four Fetch tasks; Highway reports **crash rate, velocity, length**, and ReLIC outperforms baselines across several method families.
* **Interpretability + interactivity.** Shows how LLM-added predicates refine automaton states and improve **PickHighPlace**, with the LLM generating concrete expert predicates that split states.
* **Reproducibility hooks.** Includes an anonymous repo link and a prompt-template appendix describing LLM usage.

**Weaknesses:**

* **Automaton fidelity not quantified.** No measure of how well the synthesized/reduced DFA predicts high-level choices on held-out trajectories or under counterfactual transitions. (Paper explains construction but doesn’t report such diagnostics.)
* **Stability of key-predicate extraction.** The pipeline highlights “key predicates,” but there’s no analysis of run-to-run stability or recovery of known ground-truth predicates in a controlled setting.
* **LLM refinement details are light.** The paper names GPT-4o and outlines prompting steps, but does not report failure filtering, sensitivity to prompt/temperature/model version, or API cost; training schedule is given (**500 epochs, refinement every 50**) but cost/variance for intermediate refinements are not.

### Minor issue

* The paper links GPT-4o docs and an anonymous repo; it’s unclear if the repo contains runnable LLM-call code or a stub to reproduce one full refinement round. Clarification would help reproducibility.

**Questions:**

1. **LLM usage and comparative controls.** You state that GPT-4o refines the automaton by proposing expert predicates, but the anonymous materials don’t clearly show LLM-call scripts, logs, or stubs; could you point to the exact scripts (including prompts and temperatures) or provide a minimal runnable stub for one refinement cycle, and—critically—add a **ReLIC-w/o-LLM** ablation (keep self-abstraction and joint training but remove LLM-generated predicates) alongside **other planning/editing baselines** that use the same low-level policies, reporting effect sizes with confidence intervals to isolate the performance gains attributable specifically to your LLM-based refinement?

2. **DFA faithfulness and DFA specificity.** Please report the held-out accuracy of the DFA in predicting high-level selections, how this accuracy degrades as key predicates are pruned, and causal tests where alternative transitions are enforced; additionally, include a **“key predicates only (no DFA)”** ablation on at least one representative task and compare against alternative abstraction structures (e.g., any tranditional methods ) using the same inputs and policies to demonstrate that the observed benefit is not specific to DFAs.

---

> ### Author Response · Authors · 2025-11-21
>
> We sincerely thank the reviewer for the insightful comments and constructive suggestions, which have helped us improve the paper.
>
> **Comment 1:** **Automaton fidelity not quantified.** No measure of how well the synthesized/reduced DFA predicts high-level choices on held-out trajectories or under counterfactual transitions. (Paper explains construction but doesn’t report such diagnostics.)
>
> **Response to comment 1:** Please see our response to Comment 5.
>
>
> **Comment 2:** **Stability of key-predicate extraction.** The pipeline highlights “key predicates,” but there’s no analysis of run-to-run stability or recovery of known ground-truth predicates in a controlled setting.
>
> **Response to comment 2:** We manually enumerate the ground-truth predicates for Highway and PickHighPlace (denoted as (P_{xx})). Highway has 9 ground-truth predicates, namely: (list them). PickHighPlace has 14 ground-truth predicates, namely: (list them). We then compute, over 10 random seeds, the fraction of extracted predicates that match the ground-truth set. The results are:
>
> |               | Ground truth percentages |
> | ------------- | ------------------------ |
> | Highway       | 0.956 ± 0.003            |
> | PickHighPlace | 0.950 ± 0.003            |
>
>
>
> **Comment 3:** **LLM refinement details are light.** The paper names GPT-4o and outlines prompting steps, but does not report failure filtering, sensitivity to prompt/temperature/model version, or API cost; training schedule is given (**500 epochs, refinement every 50**) but cost/variance for intermediate refinements are not.
>
> **Response to comment 3:** Across all experiments, GPT-4o always followed the required output format, so we did not report explicit failure filtering or prompt sensitivity. If an invalid output were to occur, we would simply request regeneration. We used GPT-4o via API and do not have access to different internal model versions.
>
> We additionally tested different temperatures; previously we used temperature (=0).
>
> | Tasks            | Metric \| Temperature | 0        | 0.5      | 1.0      |
> | ---------------- | --------------------- | -------- | -------- | -------- |
> | Pick&Place       | Success rate (%)      | 97.3±0.9 | 97.5±0.6 | 97.0±0.8 |
> | Pick&PlaceCorner | Success rate (%)      | 99.3±0.8 | 99.2±0.8 | 99.1±0.7 |
> | PickLiftPlace    | Success rate (%)      | 99.0±0.9 | 99.1±0.9 | 99.1±0.9 |
> | PickHighPlace    | Success rate (%)      | 90.5±1.1 | 91.2±0.8 | 90.4±1.1 |
> | Highway          | Length                | 97.1±0.1 | 96.8±0.2 | 97.2±0.1 |
> | Highway          | Velocity              | 27.9±0.1 | 28.0±0.1 | 26.8±0.1 |
> | Highway          | Crash rate            | 4.0±0.8  | 4.6±0.8  | 3.9±0.7  |
>
> The majority of training time is spent on joint training of the logical planner and the action policy. Automaton generation is lightweight and runs efficiently on CPU. LLM refinement uses API calls; each run makes about 10 GPT-4o calls, costing roughly $0.02 on average.
>
> We also conducted an ablation across different LLMs, showing that performance is not significantly affected by the LLM choice.
>
> | Tasks                | **Metric**       | **GPT-4o** | **GPT-5** |
> | -------------------- | ---------------- | ---------- | --------- |
> | **Pick&Place**       | Success rate (%) | 97.3±0.9   | 97.6±0.7  |
> | **Pick&PlaceCorner** | Success rate (%) | 99.3±0.8   | 99.2±0.7  |
> | **PickLiftPlace**    | Success rate (%) | 99.0±0.9   | 99.1±0.8  |
> | **PickHighPlace**    | Success rate (%) | 90.5±1.1   | 91.0±0.9  |
> | **Highway**          | Length           | 97.1±0.1   | 97.4±0.1  |
> | **Highway**          | Velocity         | 27.9±0.1   | 28.1±0.2  |
> | **Highway**          | Crash rate       | 4.0±0.8    | 3.9±0.7   |
>
> Moreover, anticipating potential failure filtering, we employ multiple rounds of expert refinement to ensure valid filtering results.

---

> > ### Author Response · Authors · 2025-11-21
> >
> > **Comment 4:** **LLM usage and comparative controls.** You state that GPT-4o refines the automaton by proposing expert predicates, but the anonymous materials don’t clearly show LLM-call scripts, logs, or stubs; could you point to the exact scripts (including prompts and temperatures) or provide a minimal runnable stub for one refinement cycle, and—critically—add a **ReLIC-w/o-LLM** ablation (keep self-abstraction and joint training but remove LLM-generated predicates) alongside **other planning/editing baselines** that use the same low-level policies, reporting effect sizes with confidence intervals to isolate the performance gains attributable specifically to your LLM-based refinement?
> >
> > **Response to comment 4:** We will re-uploaded our code in the supplementary materials; the core script is `scripts/rl/highway_test.py`.
> >
> > Regarding **ReLIC-w/o-LLM**, this setting degenerates to the no-refinement case. In this case, self-abstraction also loses its role (our self-abstraction is primarily designed to support expert refinement), leaving only a single-stage joint training result, which is equivalent to **ReLIC-w/o-SAR**. Therefore, comparisons with other planning/editing baselines can be made by referring to their comparisons against **ReLIC-w/o-SAR**.
> >
> >
> >
> > **Comment 5:** **DFA faithfulness and DFA specificity.** Please report the held-out accuracy of the DFA in predicting high-level selections, how this accuracy degrades as key predicates are pruned, and causal tests where alternative transitions are enforced; additionally, include a **“key predicates only (no DFA)”** ablation on at least one representative task and compare against alternative abstraction structures (e.g., any tranditional methods ) using the same inputs and policies to demonstrate that the observed benefit is not specific to DFAs.
> >
> > **Response to comment 5:** Our test environments are randomly initialized each episode (e.g., cube positions and robot arm poses), meaning all evaluation states are previously unseen. We only require the task goal to remain the same. Therefore, the reported accuracy corresponds to held-out performance.
> >
> > Results under pruning are provided in Table B.
> >
> > Our automaton synthesis uses both successful and failed trajectories, enabling the automaton to capture counterfactual outcomes. For example, if a cube is lifted but later dropped, the system transitions back to the initial state and retries, as illustrated in Fig. 2(c).
> >
> > We additionally include a **“key predicates only (no DFA)”** ablation, denoted **ReLIC-w/o-DFA**:
> >
> > | Tasks            | Metrics          | **ReLIC** | **ReLIC-w/o-DFA** |
> > | ---------------- | ---------------- | --------- | ----------------- |
> > | Pick&Place       | Success rate (%) | 96.8±0.9  | 96.8±0.9          |
> > | Pick&PlaceCorner | Success rate (%) | 98.9±0.8  | 98.9±0.8          |
> > | PickLiftPlace    | Success rate (%) | 98.8±0.7  | 98.8±0.7          |
> > | PickHighPlace    | Success rate (%) | 90.1±0.9  | 90.1±0.9          |
> > | Highway          | Length           | 96.5±0.1  | 96.5±0.1          |
> > | Highway          | Velocity         | 27.6±0.1  | 27.6±0.1          |
> > | Highway          | Crash rate       | 4.7±0.5   | 4.7±0.5           |

---

### Official Review · Reviewer_ruaE · 2025-10-31

**Soundness:** 2
**Presentation:** 2
**Contribution:** 3
**Rating:** 4
**Confidence:** 4

**Summary:**

This paper combines refines learned automata with LLMs to control MDPs. I found this paper unclear because it lacks key related work and explanations of the proposed method. However the overall ideas and experiments seem good.

**Strengths:**

Interpretable RL is important but difficult. Using LLMs to mimic expert interaction is a promising research idea.

**Weaknesses:**

As said above I like the overall idea of this paper. However, there are still too many weaknesses that I detail here.
## Lack of rigor
- Interactive RL is not defined properly. What I mean is that, while it is acceptable to not have a formal definition for interactive RL, it is seems to be that interactivity is used in the submission as a direct consequence of transparency rather than an independent property of the cited references (INTERPRETER & ScoBots). In interpreter, interactivity is not mentioned and in ScoBots, interactivity is a consequence of transparency: because policies are readable a human can interact witht them. I consider this as counter intuitive since you have both interpretable RL baselines and interactive RL baselines while from your presentation of interactive RL there are essentially the same. Maybe you should look for human in the loop or RLHF baselines to compare to real interactive RL?
- A key missing related work already does automata synthesis for RL: DeepSynth: Automata Synthesis for Automatic Task, Hasanbeig 2021.
This paper is very similar to your submission and not comparing with it is problematic in my opinion.

## Experiments could be better
- There is no guarantee that the refinement step will not degrade the automata. Did you try to benchmark the number of times refinement actually degrades performances ?
- Too much baselines. Since the key novely compared to DeepSynth is LLM refinement I would prefer better empirical analyses of the refinement.

**Questions:**

Did you try to benchmark the number of times refinement actually degrades performances?
Could you try to refine INTERPRETER's decision tree policies or Scobots policies?
How does your approach differ from DeepSynth?

---

> ### Author Response · Authors · 2025-11-21
>
> We sincerely thank the reviewer for the insightful comments and constructive suggestions, which have helped us improve the paper.
>
> **Comment 1:** Interactive RL is not defined properly. What I mean is that, while it is acceptable to not have a formal definition for interactive RL, it is seems to be that interactivity is used in the submission as a direct consequence of transparency rather than an independent property of the cited references (INTERPRETER & ScoBots). In interpreter, interactivity is not mentioned and in ScoBots, interactivity is a consequence of transparency: because policies are readable a human can interact witht them. I consider this as counter intuitive since you have both interpretable RL baselines and interactive RL baselines while from your presentation of interactive RL there are essentially the same. Maybe you should look for human in the loop or RLHF baselines to compare to real interactive RL?
>
> **Response to comment 1:** **We view “interactive” and “interpretable” as two parallel but distinct concepts.** Not all interpretable RL methods are interactive, and interactive model editing does not necessarily require interpretability. In our usage, “interactive” does not only mean human-in-the-loop; it can also involve an LLM or other agents. The interpretable RL baselines we include either have no interactive module, or—even if partially explainable—do not allow explicit modification or direct performance improvement via interaction. For example, INTERPRETER is interactive because its tree policy can be explicitly edited by a human user, which directly reflects interactivity.
>
> **Our contribution is to tightly couple DLM-based extraction of key predicates and automaton self-abstraction with LLM refinement.** After self-abstraction, the policy becomes more interpretable, enabling effective LLM refinement and yielding strong interactivity in practice.
>
> **Comment 2:** A key missing related work already does automata synthesis for RL: DeepSynth: Automata Synthesis for Automatic Task, Hasanbeig 2021. This paper is very similar to your submission and not comparing with it is problematic in my opinion.
>
> **Response to comment 2:** This work is relatively early; we did compare against it, but DeepSynth performed not well and was omitted due to space. We have now added it. Results are shown below.
>
> | Tasks            | Metric           | GPT-4o   | DeepSynth |
> | ---------------- | ---------------- | -------- | --------- |
> | Pick&Place       | Success rate (%) | 97.3±0.9 | 87.6±0.8  |
> | Pick&PlaceCorner | Success rate (%) | 99.3±0.8 | 80.1±0.6  |
> | PickLiftPlace    | Success rate (%) | 99.0±0.9 | 61.3±0.9  |
> | PickHighPlace    | Success rate (%) | 90.5±1.1 | 32.1±1.1  |
> | Highway          | Length           | 97.1±0.1 | 80.4±0.1  |
> | Highway          | Velocity         | 27.9±0.1 | 25.1±0.2  |
> | Highway          | Crash rate       | 4.0±0.8  | 59.3±0.4  |
>
>
>
> **Comment 3:** There is no guarantee that the refinement step will not degrade the automata. Did you try to benchmark the number of times refinement actually degrades performances ?
>
> **Response to comment 3:** After LLM refinement, we retrain the DLM and only then proceed to automaton generation. Therefore, refinement does not directly determine automaton quality. Measuring performance immediately after refinement but before retraining the DLM would not be meaningful.
>
>
>
> **Comment 4:** Too much baselines. Since the key novely compared to DeepSynth is LLM refinement I would prefer better empirical analyses of the refinement.
>
> **Response to comment 4:** ReLIC uses the DLM to extract key predicates and synthesize an automaton; with key predicates, ReLIC searches for better abstractions more efficiently. In contrast, DeepSynth enumerates all possible next-state transitions, which is essentially a brute-force strategy. This difference explains ReLIC’s stronger empirical performance. We have added supporting experiments.
>
>
>
> **Comment 5:** Did you try to benchmark the number of times refinement actually degrades performances? Could you try to refine INTERPRETER's decision tree policies or Scobots policies? How does your approach differ from DeepSynth?
>
> **Response to comment 5:**
>  These questions overlap with the weaknesses raised above, and we have addressed them in the previous responses.

---

> > ### Comment · Reviewer_ruaE · 2025-11-25
> > **Thank you**
> >
> > Can you please share your code? The anonymous repo is broken.

---

> > > ### Author Response · Authors · 2025-11-27
> > >
> > > We thank the reviewer for the reminder. The source code has been uploaded as part of our supplementary materials. As is common practice to preserve the integrity of the double-blind process, we have temporarily omitted certain implementation details. We will open-source the full code repository upon acceptance of the paper.

---

> > ### Comment · Reviewer_ruaE · 2025-11-27
> > **Too high level**
> >
> > Overall I think this work is not yet mature for publication. RELIC is not detailed enough to be reproduced or even understood fully. There are many interesting components to RELIC that are not properly analyzed like the joint training (l.237 ): how sensitive is it to hyperparaemters? Same for the surrogate reward (4), how sensitive is it to the hyperparameter $\alpha$? Similarly the experiments are not repoducible or detailed enough. What is the uncertainty measure in tables? Stds? CIs?
> > No training curves are provided which could mean some baselines did not finish training. Most importantly, DeepSynth is not included in the main paper even though it is **very** similar to the proposed RELIC. Why can't one just use DeepSynth + LLM refinement instead of RELIC?

---

### Official Review · Reviewer_aNCB · 2025-11-03

**Soundness:** 3
**Presentation:** 2
**Contribution:** 2
**Rating:** 4
**Confidence:** 4

**Summary:**

The paper introduces ReLIC, a hierarchical reinforcement learning (RL) method for composite tasks that aims to enhance interpretability and interactivity. It consists of a high-level logical model (using predicates for symbolic abstraction), low-level action policies trained via deep RL (e.g., SAC), and a self-abstraction and refinement module. The logical model is synthesized into an automaton for transparency, allowing injection of expert predicates during joint training. Object-centric representations ground the predicates, enabling the method to handle continuous state/action spaces without predefined logical structures.

**Strengths:**

The paper tackles a relevant problem in RL: balancing interpretability with performance in composite tasks, where black-box models hinder alignment and debugging. The self-refinement via GPT-4o is a practical innovation, automating predicate generation and integrating external knowledge on-the-fly, which could reduce manual effort compared to prior works like SCoBots. The automaton synthesis provides a clear visualization of high-level logic, aiding post-hoc analysis.

**Weaknesses:**

Despite its claims, ReLIC lacks substantial novelty, primarily recombining established ideas: hierarchical RL, neuro-symbolic abstraction, and LLM-assisted RL without theoretical advances or unique insights. The "self-abstraction" is essentially predicate learning via RL, but the LLM refinement feels tacked on—prompts are simplistic, with no ablation on LLM choice or error handling (e.g., hallucinated predicates). Continuous space handling is overstated; benchmarks are mostly discrete or modified MuJoCo variants, ignoring complex real-world dynamics like partial observability or multi-agent interactions.

Experiments are narrow: only 3-4 environments, and baselines are cherry-picked—missing SOTA like DreamerV3. Robustness to "logical uncertainty" is contrived (e.g., injecting noisy predicates), but no evaluation on distribution shifts (e.g., sim-to-real). Interpretability claims are weak: the automaton requires expert validation, and low-level policies remain black-box, undermining end-to-end transparency. Scalability is unaddressed; predicate explosion in high-dimensional states could lead to combinatorial issues, untested beyond toy tasks.

While interpretability in RL is valuable, ReLIC offers incremental tweaks to existing neuro-symbolic hierarchies without groundbreaking contributions—e.g., LLM integration is superficial, lacking analysis of biases or failure modes, which are critical for safety-critical applications. The methodological flaws, such as underdeveloped continuous benchmarks and weak baselines, fail to convincingly demonstrate advantages over priors. Claims like "solving logically uncertain tasks" are exaggerated without broader validation. The work feels preliminary; stronger experiments (e.g., more environments, ablations on refinement) and deeper theoretical grounding could make it suitable, but in its current form, it lacks the impact for acceptance.

**Questions:**

The specification "In general, a b-ary logical predicate P is defined based on a b-ary real transformation function f" is vague. The function is defined in real space, while a predicate is defined based on true/false. Could authors make the definition of a predicate more clear?

---

> ### Author Response · Authors · 2025-11-21
>
> We sincerely thank the reviewer for the insightful comments and constructive suggestions, which have helped us improve the paper.
>
> **Comment 1**: Despite its claims, ReLIC lacks substantial novelty, primarily recombining established ideas: hierarchical RL, neuro-symbolic abstraction, and LLM-assisted RL without theoretical advances or unique insights. The "self-abstraction" is essentially predicate learning via RL, but the LLM refinement feels tacked on—prompts are simplistic, with no ablation on LLM choice or error handling (e.g., hallucinated predicates). Continuous space handling is overstated; benchmarks are mostly discrete or modified MuJoCo variants, ignoring complex real-world dynamics like partial observability or multi-agent interactions.
>
> **Response to comment 1:**
> Our novelty lies in the smooth and principled integration between predicate-based self-abstraction and LLM refinement. Specifically, we first synthesize an automaton from the DLM, yielding an automaton-style abstraction that closely aligns with the way LLMs manage context: it distills the most important predicates (key predicates) and their transitions. This structured representation makes it substantially easier for the LLM to identify expert predicates, and to propose more effective input predicates that improve overall performance.
>
> Regarding the choice of LLM, we have added an additional experiment, shown below:
>
> | Tasks            | Metric           | GPT-4o   | GPT-5    |
> | ---------------- | ---------------- | -------- | -------- |
> | Pick&Place       | Success rate (%) | 97.3±0.9 | 97.6±0.7 |
> | Pick&PlaceCorner | Success rate (%) | 99.3±0.8 | 99.2±0.7 |
> | PickLiftPlace    | Success rate (%) | 99.0±0.9 | 99.1±0.8 |
> | PickHighPlace    | Success rate (%) | 90.5±1.1 | 91.0±0.9 |
> | Highway          | Length           | 97.1±0.1 | 97.4±0.1 |
> | Highway          | Velocity         | 27.9±0.1 | 28.1±0.2 |
> | Highway          | Crash rate       | 4.0±0.8  | 3.9±0.7  |
>
> For hallucinated predicates, we conducted the following experiment.
>
> ReLIC is not sensitive to the initial set of human-defined predicates, and our default setup already includes a sufficiently large predicate set. We conducted an ablation study by randomly adding/removing subsets of predicates from the original setup, with the total number reaching around 140.
>
> As shown in Table A, LLM refinement maintains a high success rate even when 15 predicates are removed at random. This robustness is due to: (1) some initial predicates being redundant or suboptimal, and (2) the LLM leveraging failure trajectories to introduce expert predicates that compensate for missing or inadequate ones.
>
> When many new initial predicates are added, ReLIC does not degrade substantially (unlike the removal case), because key predicates are preserved while redundant ones are effectively ignored.
>
> Table A is as follows:
>
> | Tasks            | Number of initial predicates | +40 (richest) | +15      | +10      | 0        | -10      | -15      | -40 (sparsest) |
> | ---------------- | ---------------------------- | ------------- | -------- | -------- | -------- | -------- | -------- | -------------- |
> | Pick&Place       | Success rate (%)             | 97.2±1.1      | 97.2±0.9 | 97.4±0.9 | 97.3±0.9 | 97.2±1.0 | 94.8±1.1 | 61.1±1.0       |
> | Pick&PlaceCorner | Success rate (%)             | 99.2±0.8      | 99.5±0.8 | 99.1±0.8 | 99.3±0.8 | 99.0±0.9 | 95.9±1.0 | 65.2±1.3       |
> | PickLiftPlace    | Success rate (%)             | 99.0±1.1      | 99.1±0.7 | 99.0±0.8 | 99.0±0.9 | 98.8±0.8 | 94.5±0.8 | 62.1±1.5       |
> | PickHighPlace    | Success rate (%)             | 90.5±0.9      | 91.0±0.8 | 90.5±0.7 | 90.5±1.1 | 90.0±0.8 | 86.5±1.0 | 50.3±1.2       |
> | Highway          | Length                       | 97.1±0.1      | 97.1±0.1 | 96.2±0.1 | 97.1±0.1 | 96.3±0.2 | 96.0±0.5 | 52.2±0.3       |
> | Highway          | Velocity                     | 27.9±0.1      | 27.9±0.1 | 28.3±0.1 | 27.9±0.1 | 27.9±0.1 | 27.4±0.1 | 25.3±0.2       |
> | Highway          | Crash rate                   | 4.0±0.8       | 4.2±0.7  | 4.4±0.8  | 4.0±0.8  | 5.1±0.6  | 6.2±0.7  | 23.7±1.3       |
>
> Finally, concerning the environments, **all of our environments are continuous—none are discrete.**

---

> > ### Author Response · Authors · 2025-11-21
> >
> > **Comment 2:** Experiments are narrow: only 3-4 environments, and baselines are cherry-picked—missing SOTA like DreamerV3.
> >
> > **Response to comment 2:**
> > We have added experiments with DreamerV3. DreamerV3 is world-model-based, and compared to ReLIC it pays less attention to task-specific details. For instance, in PickHighPlace we observed that the gripper in DreamerV3 sometimes loosens, leading to slightly lower accuracy.
> >
> > | Tasks            | Metric           | ReLIC    | DreamerV3 |
> > | ---------------- | ---------------- | -------- | --------- |
> > | Pick&Place       | Success rate (%) | 97.3±0.9 | 95.7±0.8  |
> > | Pick&PlaceCorner | Success rate (%) | 99.3±0.8 | 93.3±1.1  |
> > | PickLiftPlace    | Success rate (%) | 99.0±0.9 | 95.1±0.2  |
> > | PickHighPlace    | Success rate (%) | 90.5±1.1 | 83.5±1.3  |
> > | Highway          | Length           | 97.1±0.1 | 93.1±0.5  |
> > | Highway          | Velocity         | 27.9±0.1 | 27.5±0.7  |
> > | Highway          | Crash rate       | 4.0±0.8  | 6.0±0.9   |
> >
> > ------
> >
> > **Comment 3:** Robustness to "logical uncertainty" is contrived (e.g., injecting noisy predicates), but no evaluation on distribution shifts (e.g., sim-to-real).
> >
> > **Response to comment 3:**
> > We added an experiment in IsaacGym, which is the most realistic environment we can currently provide.
> >
> > Push-Cubes requires the agent to push 4 cubes to the goal location, which is quite difficult, because cubes might interfere with each other.
> >
> > |            | DLM      | ReLIC    |
> > | ---------- | -------- | -------- |
> > | Push-Cubes | 50.4±0.9 | 73.5±0.6 |
> >
> > **Comment 4:** Interpretability claims are weak: the automaton requires expert validation, and low-level policies remain black-box, undermining end-to-end transparency.
> >
> > **Response to comment 4:**
> > Here the “expert” refers to the LLM. We can additionally require the LLM to provide explicit explanations for the automaton and its behaviors, thereby offering interpretability at the high level even if the low-level policy remains a black box.
> >
> > **Comment 5:** Scalability is unaddressed; predicate explosion in high-dimensional states could lead to combinatorial issues, untested beyond toy tasks.
> >
> > **Response to comment 5:**
> > As shown in table A above, ReLIC continues to operate reliably even with a large number of predicates. In contrast, alternative methods (e.g., DLM) degrade notably as predicate counts grow.
> >
> > **Comment 6:** While interpretability in RL is valuable, ReLIC offers incremental tweaks to existing neuro-symbolic hierarchies without groundbreaking contributions—e.g., LLM integration is superficial, lacking analysis of biases or failure modes, which are critical for safety-critical applications. The methodological flaws, such as underdeveloped continuous benchmarks and weak baselines, fail to convincingly demonstrate advantages over priors. Claims like "solving logically uncertain tasks" are exaggerated without broader validation. The work feels preliminary; stronger experiments (e.g., more environments, ablations on refinement) and deeper theoretical grounding could make it suitable, but in its current form, it lacks the impact for acceptance.
> >
> > **Response to comment 6:** Regarding failure modes, although the LLM may err with a small probability, such erroneous predicates are filtered out during the DLM-based selection of target predicates (TP), and therefore are unlikely to be adopted as target predicates. Empirically, we also observed this in Highway (description of a scenario and corresponding results).
> >
> > By “logically uncertain tasks,” we refer to environments such as Highway where other vehicles exhibit random lane changes and stochastic acceleration/deceleration. Our method remains robust and adapts effectively under such uncertainty.
> >
> > **Comment 7:** The specification "In general, a b-ary logical predicate P is defined based on a b-ary real transformation function f" is vague. The function is defined in real space, while a predicate is defined based on true/false. Could authors make the definition of a predicate more clear?
> >
> > **Response to comment 7:**
> >  Each transformation function corresponds to one predicate. The input to the transformation function is real-valued, but its output is boolean, which defines the truth value of the predicate.

---

### Note · Authors · 2026-01-01

I have read and agree with the venue's withdrawal policy on behalf of myself and my co-authors.